# GENERATIVE MODELING HELPS WEAK SUPERVISION (AND VICE VERSA)

**Benedikt Boecking**[*] **Nicholas Roberts**[†] **Willie Neiswanger**[‡]
**Stefano Ermon**[‡] **Frederic Sala**[†] **Artur Dubrawski**[*]

[*]Carnegie Mellon University   [†]University of Wisconsin   [‡]Stanford University

## ABSTRACT

Many promising applications of supervised machine learning face hurdles in the acquisition of labeled data in sufficient quantity and quality, creating an expensive bottleneck. To overcome such limitations, techniques that do not depend on ground truth labels have been studied, including weak supervision and generative modeling. While these techniques would seem to be usable in concert, improving one another, how to build an interface between them is not well-understood. In this work, we propose a model fusing programmatic weak supervision and generative adversarial networks and provide theoretical justification motivating this fusion. The proposed approach captures discrete latent variables in the data alongside the weak supervision derived label estimate. Alignment of the two allows for better modeling of sample-dependent accuracies of the weak supervision sources, improving the estimate of unobserved labels. It is the first approach to enable data augmentation through weakly supervised synthetic images and pseudolabels. Additionally, its learned latent variables can be inspected qualitatively. The model outperforms baseline weak supervision label models on a number of multiclass image classification datasets, improves the quality of generated images, and further improves end-model performance through data augmentation with synthetic samples.

## 1 INTRODUCTION

How can we get the most out of data when we do not have ground truth labels? Two prominent paradigms operate in this setting. First, programmatic *weak supervision* frameworks use weak sources of training signal to train downstream supervised models, without needing access to ground-truth labels (Riedel et al., 2010; Ratner et al., 2016; Dehghani et al., 2017; Lang & Poon, 2021). Second, *generative models* enable learning data distributions which can benefit downstream tasks, e.g. via data augmentation or representation learning, in particular when learning latent factors of variation (Higgins et al., 2018; Locatello et al., 2019; Hu et al., 2019). Intuitively, these two paradigms should complement each other, as each can be thought of as a different approach to extracting structure from unlabeled data. However, to date there is no simple way to combine them.

Fusing generative models with weak supervision holds substantial promise. For example, it could yield large reductions in data acquisition costs for training complex models. Programmatic weak supervision replaces the need for manual annotations by applying so-called labeling functions to unlabeled data, producing weak labels that are combined into a pseudolabel for each sample. This leaves the majority of the acquisition budget to be spent on unlabeled data, and here generative modeling can reduce the number of real-world samples that need to be collected. Similarly, information about the data distribution contained in weak label sources may improve generative models, reducing the need to acquire large volumes of samples to increase generative performance and model discrete structure. Additionally, learning with weak labels may enable targeted data augmentation, allowing for class-conditional sample generation despite not having access to ground truth.

---

[*]boecking@cmu.edu, awd@cmu.edu
[†]nick11roberts@cs.wisc.edu, fredsala@cs.wisc.edu
[‡]neiswanger@cs.stanford.edu, ermon@cs.stanford.edu

The main technical challenge is to build an *interface* between the core models used in the two approaches. Generative adversarial networks (GANs) (Goodfellow et al., 2014), which we focus on in this work, have at least a generator and a discriminator, and frequently additional auxiliary models, such as those that learn to disentangle latent factors of variation (Chen et al., 2016). In programmatic weak supervision, the *label model* is the main focus. It is necessary to develop an interface that aligns the structures learned from the unlabeled data by the various components.

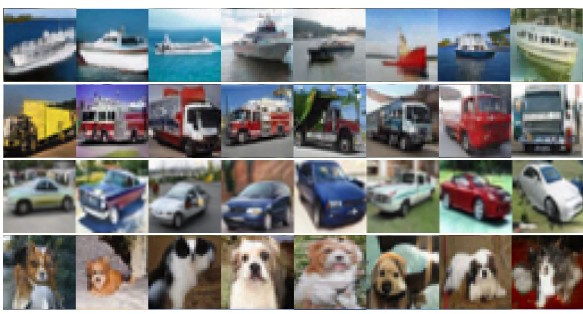

Figure 1: Class-conditional image generation by the proposed WSGAN based on a ***weakly supervised*** CIFAR10 subset with 30k samples. Here, WSGAN uses a StyleGAN2 base architecture and we keep the discrete code in each row fixed.

We introduce *weakly-supervised GAN (WS-GAN)*, a simple yet powerful fusion of weak supervision and GANs visualized in Fig. 2, and we provide a theoretical justification that motivates the expected gains from this fusion. Our WSGAN approach is related to the unsupervised InfoGAN (Chen et al., 2016) generative model, and also inspired by encoder-based label models as in (Cachay et al., 2021). These techniques expose structure in the data, and our approach ensures alignment between the resulting variables by learning projections between them. The proposed WSGAN offers a number of benefits, including:

- **Improved weak supervision:** We obtain better-quality pseudolabels via WSGAN's label model, yielding consistent improvements in pseudolabel accuracy up to 6% over established programmatic weak supervision techniques such as Snorkel (Ratner et al., 2020).
- **Improved generative modeling:** Weak supervision provides information about unobserved labels which can be used to obtain better disentangled latent variables, thus improving the model's generative performance. Over 6 datasets, our WSGAN approach improves image generation by an average of 5.8 FID points versus InfoGAN. We conduct architecture ablations and show that the proposed approach can be integrated into state-of-the-art GAN architectures such as *StyleGAN* (Karras et al., 2019) (see Fig. 1), achieving state-of-the-art image generation quality.
- **Data augmentation via synthetic samples:** WSGAN can generate samples and corresponding label estimates for data augmentation (e.g. Fig. 10), providing improvements of downstream classifier accuracy of up to 3.9% in our experiments. The trained WSGAN can produce label estimates even for samples, real or fake, that have no weak supervision signal available.

## 2 BACKGROUND

We propose to fuse weak supervision with generative modeling to the benefit of both techniques, and first provide a brief overview. A broader review of related work is presented in Section 5.

**Weak Supervision**    Weak supervision methods that use multiple sources of imperfect and partial labels (Ratner et al., 2016; 2020; Cachay et al., 2021), sometimes referred to as *programmatic weak supervision*, seek to replace manual labeling for the construction of large labeled datasets. Instead, users define multiple weak label sources that can be applied automatically to the unlabeled dataset. Such sources can be heuristics, knowledge base look-ups, off-the-shelf models, and more. The technical challenge is to combine the source votes into a high-quality pseudolabel via a *label model*. This requires estimating the errors and dependencies between sources and using them to compute a posterior label distribution. Prior work has considered various choices for the label model, most of which only take the weak source outputs into account. A review can be found in Zhang et al. (2021; 2022). Instead, our label model produces sample dependent accuracy estimates for the weak sources based on the features of the data, similar to Cachay et al. (2021).

**Generative Models and GANs**    Generative models are used to model and sample from complex distributions. Among the most popular such models are generative adversarial networks (GANs) (Goodfellow et al., 2014). GANs consist of a generator and discriminator model that play a minimax game against each other. Our approach builds off InfoGAN (Chen et al., 2016), which adds an auxiliary inference component to learn disentangled representations via a set of latent factors of variation. We hypothesize that connecting such discrete latent variables to the label model should yield benefits for both weak supervision and generative modeling.

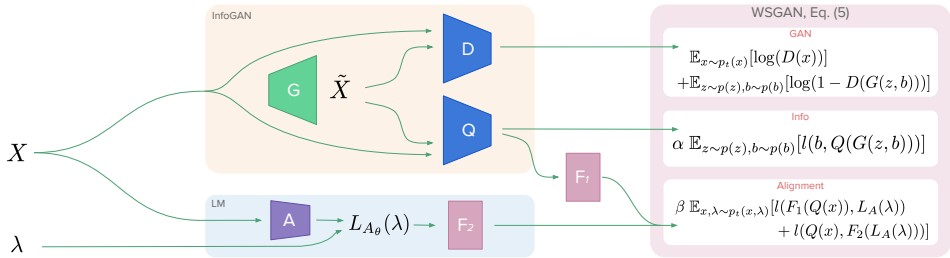

Figure 2: The proposed WSGAN models discrete latent variables in $X$ via a network $Q$, while learning a generator $G$ and discriminator $D$. A label model $L$ uses weak supervision votes $\lambda$ and weights estimated by $A$ to produce pseudolabels. WSGAN aligns the pseudolabels with the discrete structure learned by $Q$.

## 3   THE WSGAN MODEL

We first describe our proposed weakly-supervised GAN (WSGAN) model, visualized in Fig. 2, and then provide theoretical justification for the model fusion. We work with $n$ unlabeled samples $X \in \mathcal{X} \subseteq \mathbb{R}^d$ drawn from a distribution $\mathcal{D}_X$. We want to achieve two goals with the samples $X$. First, in generative modeling, we approximate $\mathcal{D}_X$ with a model that can be used to produce high-fidelity synthetic samples. Second, in supervised learning, we wish to use $X$ to predict labels $Y \in \{1, 2, \ldots, C\}$, where $(X, Y)$ is drawn from a distribution whose marginal distribution is $\mathcal{D}_X$. However, in the weak supervision setting, we do not observe $Y$. Instead, we observe $m$ *labeling functions* (LFs) $\Lambda \in \{0, \ldots, C\}^{n \times m}$ that provide imperfect estimates of $Y$ for a subset of the samples. These LFs vote on a sample $x_i$ to produce an estimate of the label $\lambda_j(x_i) \in \{1, \ldots, C\}$ or abstain (i.e. no vote) with 0. The goal is to combine the $m$ LF estimates into a pseudolabel $\hat{Y}$ that can be used to train a supervised model (Ratner et al., 2016). While weak supervision and generative modeling function over a number of modalities, this work focuses on images. Note that LF construction for image tasks is more challenging than for text tasks (cf. Section 5).

### 3.1   PROPOSED METHOD

To improve generative performance and the weak supervision-based pseudolabels, we propose a model that consists of a number of components. Because we wish to ensure that our component models benefit each other, our architecture aims for the following characteristics: (I) A generative model component that learns discrete latent factors of variation from data and exposes these externally, (II) a weak supervision label model component that makes predictions of the unobserved label by aggregating the weak supervision votes, using sample-dependent weights, (III) a set of *interface* models that connect the components. Our design choices are made to satisfy those goals.

**GAN Architecture**   We write $G$ for the generator; its goal is to learn a mapping to the image space based on input consisting of samples $z$ from a noise distribution $p_Z(z)$ along with a set of latent factors of variation $b \sim p(b)$, following the ideas introduced in InfoGAN (Chen et al., 2016). Because we are targeting a classification setting, we restrict ourselves to discrete $b$. The output of $G$ are samples $x$; these are consumed by a discriminative model $D$, which estimates the probability that a sample came from the training distribution rather than $G$. Furthermore, we define an auxiliary model $Q$ which learns to map from a sample $x$ to the discrete latent code $b$. We denote the standard GAN objective by V(D,G), and the InfoGAN objective by IV(D,G,Q) (Chen et al., 2016):

$$\min_G \max_D V(D, G) \quad = \mathbb{E}_{x \sim \mathcal{D}_X}\left[\log(D(x))\right] + \mathbb{E}_{z \sim p(z), b \sim p(b)}\left[\log(1 - D(G(z, b)))\right], \quad (1)$$

$$\min_{G,Q} \max_D IV(D, G, Q) = V(D, G) + \ \alpha \ \mathbb{E}_{z \sim p(z), b \sim p(b)}\left[l(b, Q(G(z, b)))\right], \quad (2)$$

where $l$ is an appropriate loss function, such as cross entropy, and $\alpha$ is a trade-off parameter. Equation 2 aims to maximize the mutual information between generated images and $b$, while $G$ continues to fool the discriminator $D$, leading to the discovery of latent factors of variation.

**Weak Supervision Label Model**   The purpose of the *label model* is to encode relationships between the LFs $\lambda$ and the unobserved label $y$, enabling us to produce an informed estimate of $y$. In prior work, the model is often a factor graph (Ratner et al., 2016; 2019; Fu et al., 2020; Zhang et al., 2022) with potentials $\phi_j(\lambda_j(x), y)$ and $\phi_{j,k}(\lambda_j(x), \lambda_k(x))$ capturing the degree of agreement between an LF $\lambda_j$ and $y$ or correlations between two LFs $\lambda_j$ and $\lambda_k$. We define the accuracy potentials

$\phi_j(\lambda_j, y) \triangleq \mathbb{1}\{\lambda_j = y\}$ as in related work. Each potential $\phi_j$ is associated with an accuracy parameter $\theta_j$. Once we obtain estimates of $\theta_j$, we can predict $y$ from the LFs $\lambda$ via

$$L_\theta(\lambda)_k = \frac{\exp(\sum_{j=1}^m \theta_j \phi_j(\lambda_j(x), k))}{\sum_{\tilde{y} \in \mathcal{Y}} \exp(\sum_{j=1}^m \theta_j \phi_j(\lambda_j(x), \tilde{y}))} \ , \ \ \forall \, k \in \{1, \dots, C\}.$$

This is a softmax over the weighted votes of all LFs, which derives from the factor graph introduced in Ratner et al. (2016). Note that related work only models the LF outputs to learn $\theta$, ignoring any additional information in the features $x$. However, the structure in the input data $x$ is crucial to our fusion. For this reason, we define a modified label model predictor in the spirit of Cachay et al. (2021). It has *local* accuracy parameters (sample-dependent values encoding the accuracy of each $\lambda_j$) via an accuracy parameter encoder $A(x) : \mathbb{R}^d \to \mathbb{R}_+^m$. This variant is given by:

$$L_{A_\theta}(\lambda)_k = \frac{\exp(\sum_{j=1}^m A(x)_j \phi_j(\lambda_j(x), k))}{\sum_{\tilde{y} \in \mathcal{Y}} \exp(\sum_{j=1}^m A(x)_j \phi_j(\lambda_j(x), \tilde{y}))} \ , \ \ \forall \, k \in \{1, \dots, C\}, \tag{3}$$

a softmax over the LF votes by class, weighted by the accuracy encoder output. Note that, while $A(x)$ allows for finer-grained adjustments of the label estimate $\hat{Y}$, the estimate is still anchored in the votes of LFs which represent strong domain knowledge and are assumed to be better than random.

**Learning the Label Model** The technical challenge of weak supervision is to learn the parameters of the label model (such as $\theta_j$ above) without observing $y$. Existing approaches find parameters under a label model that (i) best explain the LF votes while (ii) satisfying conditionally independent relationships (Ratner et al., 2016; 2019; Fu et al., 2020). The features $x$ are ignored; it is assumed that all information about $y$ is present in the LF outputs. Instead, we promote cooperation between our models by ensuring that *the best label model is the one which agrees with the discrete structure that the GAN can learn, and vice versa*. The intuition is that, as each of the generative and label models learn useful information from data, this information can—if aligned correctly—be shared to help teach the other model. To this end, note that the sampled variable $b$ can only be observed for generated images, not for real images. Nonetheless, $Q$ can be applied to real-world samples to obtain a prediction of the latent $b$. Crucially, in the weak supervision setting we observe the LF outputs, enabling us derive a label estimate for each real image $L_{A_\theta}(\lambda) = \hat{Y}$, which can be aligned with the predicted code to guide $Q$ on real data, and vice versa.

**Interface Models and Overall WSGAN Objective** We introduce the following *interface* models to map between the estimates of $b$ and $y$. Let $F_1 : [0,1]^C \to [0,1]^C$ and $F_2 : [0,1]^C \to [0,1]^C$. An effective choice for $F_1$ and $F_2$ are linear models with a softmax activation function. To achieve agreement between the latent structure discovered by the GAN's auxiliary model $Q$ as well as by the label model $L_{A_\theta}$ via the LFs, we introduce the following overall objective, ensuring that a mapping exists between the latent structures on the real images in the training data:

$$\min_{G,Q,A,F_1,F_2} \max_D IV(D, G, Q) + \beta \, \mathbb{E}_{x, \lambda \sim \mathcal{D}_{X,\Lambda}} [l(F_1(Q(x)), L_A(\lambda)) + l(Q(x), F_2(L_A(\lambda)))], \tag{4}$$

with hyperparameter $\beta$ and loss function $l$, such as the cross entropy. Pseudocode for the added loss term can be found in Algorithm 1. In our implementation, as common in related GAN work, we let $D, Q$ and $A$ share convolutional layers and define distinct prediction heads for each. For $L_A$ we detach the features from the computation graph before passing them to a multilayer perceptron (MLP), followed by a sigmoid activation function. Thus, the WSGAN method only adds a small number of additional parameters compared to a basic GAN or InfoGAN.

**Improving Alignment** Initializing the label model $L_A$ such that it produces equal weights for all LFs results in a strong baseline estimate of $\hat{Y}$, as users build LFs to be better than random. Initializing $L_{A_\theta}$ in this way, it can act as a teacher in the beginning and guide $Q$ towards the discrete structure encoded in the LFs. We find that adding a decaying penalty term that encourages equal label model weights in early epochs–while not necessary to achieve good performance–almost always improves latent label estimates. Let $i \geq 0$ denote the current epoch. We propose to add the following linearly decaying penalty term for an encoder $A$ that uses a sigmoid activation function: $C/(i \times \gamma + 1)||A(x) - \vec{1} \times 0.5||_2^2$, where $\gamma$ is a decay parameter. In our experiments we set $\gamma = 1.5$.

**Augmenting the Weak Supervision Pipeline with Synthetic Data**    Given a WSGAN model trained according to Eq. 4, we can generate images via $G$ to obtain unlabeled synthetic samples $\tilde{x}$. To obtain pseudolabels for these images we have at least one and sometimes two options. When LFs can be applied to synthetic images, we can obtain their votes $\lambda(\tilde{x}) = \tilde{\lambda}$ and apply our WSGAN label model $L_A(\tilde{\lambda})$. However, in many practical applications of weak supervision, some LFs are not applied to images directly, but rather to metadata or an auxiliary modality such as text (cf. Section 5). With WSGAN, we can obtain pseudolabels via $\hat{y} = F_1(Q(\tilde{x}))$ for samples that have no LF votes, using the trained WSGAN components $Q$ and $F_1$, in essence transferring knowledge from $Q$ to the end model. Note that the quality of these synthetic pseudolabels hinges on the performance of $Q$, which can conceivably improve with the supply of weakly supervised as well as entirely unlabeled data.

## 3.2    THEORETICAL JUSTIFICATION

In this section, we provide theoretical results that suggest that there is a provable benefit to combining weak supervision and generative modeling. In particular, we provide two theoretical claims justifying why *weak supervision should help generative modeling (and vice versa)*: (1) generative models help weak supervision via a generalization bound on downstream classification and (2) weak supervision improves a multiplicative approximation bound on the loss for a conditional GAN using the unobserved true labels—namely, we extend the theoretical setup and noisy channel model of the Robust Conditional GAN (RCGAN) (Thekumparampil et al., 2018). Formal statements and proofs of these claims can be found in Appendix F.

**Claim (1)**    Assume that we have $n_1$ unlabeled real examples where our label model fails to produce labels, i.e. all LFs abstain on these $n_1$ points. This is a typical issue in weak supervision, as sources often only vote on a small proportion of points. We then sample enough synthetic examples from our generative model such that we obtain $n_2$ synthetic examples for which our label model *does* produce labels; this enables training of a downstream classifier on synthetic examples alone with the following generalization bound:

$$\sup_{f \in \mathcal{F}} |\hat{\mathbb{R}}_{\hat{\mathcal{D}}}(f) - \mathbb{R}_{\mathcal{D}}(f)| \leq 2\mathfrak{R} + \sqrt{\frac{\log(1/\delta)}{2n_2}} + B_\ell G^{\frac{1}{2}} + B_\ell \sqrt{2} \exp(-m\alpha^2),$$

where $\mathfrak{R}$ is the Rademacher complexity of the function class. The first two terms are standard. The third term is the penalty due to generative model usage; any generative model estimation result for total variation distance can be plugged in. For example, for estimating a mixture of Gaussians, $G = (4c_G k d^2/n_1)^{1/2}$ which depends on the number of mixture components $k$ and dimension $d$. The last term is the penalty from weak supervision with $m$ LFs whose accuracy is $\alpha$ better than chance; this implies that generated samples can help weak supervision generalize when true samples cannot.

**Claim (2)**    Noisy labels from majority vote improve the multiplicative bound on the RCGAN loss given in Theorem 2 of Thekumparampil et al. (2018). Let $P$ and $Q$ be two distributions over $\mathcal{X} \times \{0, 1\}$ and let $\widetilde{P}_{\text{MV}}$ and $\widetilde{Q}_{\text{MV}}$ be the corresponding distributions with noisy labels generated by majority vote over $m$ LFs. Let $d_{\mathcal{F}}(\widetilde{P}_{\text{MV}}, \widetilde{Q}_{\text{MV}})$ be the RCGAN loss with noisy labels generated by majority vote and let $\epsilon_\lambda$ be the mean error of each of the $m$ LFs. Using majority vote with $m \geq 0.5 \log(1/\epsilon_\lambda)/\left(\frac{1}{2} - \epsilon_\lambda\right)^2$ LFs, we obtain an exponentially tighter multiplicative bound on the noiseless RCGAN loss:

$$d_{\mathcal{F}}(\widetilde{P}_{\text{MV}}, \widetilde{Q}_{\text{MV}}) \leq d_{\mathcal{F}}(P, Q) \leq \left(1 - 2\exp\left(-2m\left(\frac{1}{2} - \epsilon_\lambda\right)^2\right)\right)^{-1} d_{\mathcal{F}}(\widetilde{P}_{\text{MV}}, \widetilde{Q}_{\text{MV}})$$

$$\leq (1 - 2\epsilon_\lambda)^{-1} d_{\mathcal{F}}(\widetilde{P}_{\text{MV}}, \widetilde{Q}_{\text{MV}}).$$

This means that weak supervision can help an RCGAN more-accurately learn the true joint distribution, even when the true labels are unobserved. The full analysis is provided in Appendix F.

## 4    EXPERIMENTS

Our experiments on multiple image datasets show that the proposed WSGAN approach is able to take advantage of the discrete latent structure it discovers in the images, leading to better label model performance compared to prior work. The results also indicate that weak supervision as used by WSGAN can improve image generation performance. In the spirit of democratizing AI, we aim to

Table 1: Datasets and labeling function (LF) characteristics used to evaluate the proposed WSGAN. Acc denotes accuracy, and Coverage denotes the proportion of samples where the LF does not abstain.

| Dataset | #Classes | #LFs | #Samples | Mean LF Acc | Min LF Acc | Max LF Acc | Mean Coverage | LF Type |
|---|---|---|---|---|---|---|---|---|
| AwA2 -A | 10 | 29 | 6726 | 0.504 | 0.053 | 0.850 | 0.104 | Attribute heuristics |
| AwA2 -B | 10 | 32 | 6726 | 0.548 | 0.116 | 0.783 | 0.131 | Attribute heuristics |
| DomainNet | 10 | 4 | 6369 | 0.493 | 0.416 | 0.684 | 1.000 | Domain transfer |
| MNIST | 10 | 29 | 30000 | 0.791 | 0.564 | 0.931 | 0.047 | SSL, finetuning |
| FashionMNIST | 10 | 23 | 30000 | 0.773 | 0.542 | 0.949 | 0.047 | SSL, finetuning |
| GTSRB | 43 | 100 | 22640 | 0.837 | 0.609 | 0.949 | 0.007 | SSL, finetuning |
| CIFAR10-A | 10 | 20 | 30000 | 0.773 | 0.624 | 0.896 | 0.061 | Synthetic |
| CIFAR10-B | 10 | 20 | 30000 | 0.736 | 0.531 | 0.912 | 0.042 | SSL, finetuning |

keep the complexity of our experiments manageable, to ensure accessible reproducibility. Therefore, we conduct our main experiments with a simple DCGAN base architecture. As an ablation, we also adapt StyleGAN2-ADA (Karras et al., 2020) to WSGAN, showing that the proposed method can be integrated with other GAN architectures to achieve state-of-the-art image generation and label model performance. Please see the Appendix for additional details and experiments as well as a link to code.

## 4.1 SETUP

**Datasets** Table 1 shows key characteristics of the datasets used in our experiments, including information about the different LF sets. We conduct our main experiments with the Animals with Attributes 2 (AwA2) (Mazzetto et al., 2021a), DomainNet (Peng et al., 2019), the German Traffic Sign Recognition Benchmark (GTSRB) (Stallkamp et al., 2012), and CIFAR10 (Krizhevsky, 2009) color image datasets, as well as with the gray-scale MNIST (LeCun et al., 1998) and FashionMNIST (Xiao et al., 2017) datasets. We use a variety of types of weak supervision sources for these datasets (see Appendix B for more dataset details). The LF types we cover are:

- *Domain transfer*: classifiers are trained on images in source domains (e.g. paintings), and the trained classifiers are then applied to images in a target domain (e.g. real images) to obtain weak labels. This LF type is used in our DomainNet experiments, following Mazzetto et al. (2021a).
- *Attribute heuristics*: we use these LFS in our AwA2 experiments. Attribute classifiers are trained on a number of seen classes of animals. Given these weak attribute predictions, we use the known attribute relations and a small amount of validation data to train shallow decision trees to produce weak labels for a set of unseen classes of animals.
- *SSL-based*: using image features learned on ImageNET with SimCLR (Chen et al., 2020), we fine-tune shallow multilayer perceptron classifiers on small sets of held-out data to produce weak labels for our datasets.
- *Synthetic*: these simulated LFs, used in some of our CIFAR10 experiments, are unipolar LFs based on the corrupted true class label. To this end, random errors are introduced to the class label to achieve a sampled target accuracy and propensity.

**Models** We study two versions of the proposed WSGAN model: (1) *WSGAN-Encoder*, which uses an *accuracy parameter encoder* $A(x)$, that takes in an image $x$ and outputs an accuracy weight vector for the label model. (2) *WSGAN-Vector*, a baseline which learns a *parameter vector* that is used to weigh LF votes and is not sample-dependent.

For our main experiments, $G, D$ follow a simple DCGAN (Radford et al., 2015) design. All networks are trained from scratch and we use the same hyperparameter settings in all experiments. For our architecture ablation, we adapt StyleGAN2-ADA (Karras et al., 2020) to create StyleWSGAN. See Appendix A for implementation details and parameter settings.

We compare WSGAN to the following *label model* approaches: (I) Snorkel (Ratner et al., 2016; 2020): a probabilistic graphical model that estimates LF parameters by maximizing the marginal likelihood using observed LFs. (II) Dawid-Skene (Dawid & Skene, 1979): a model motivated by the crowdsourcing setting. The model, fit using expectation maximization, assumes that error statistics of sources are the same across classes and that errors are equiprobable independent of the true class. (III) Snorkel MeTaL (Ratner et al., 2019): a Markov random field (MRF) model similar to Snorkel which uses a technique to complete the inverse covariance matrix of the MRF during model fitting, and also allows for modeling multi-task weak supervision. (IV) FlyingSquid (FS) (Fu et al., 2020): based on a label model similar to Snorkel, FS provides a closed form solution by augmenting it to set up a binary Ising model, enabling scalable model fitting. (V) Majority Vote (MV): A standard scheme that uses the most popular LF output as the estimate of the true label.

Table 2: Average posterior accuracy of various label models on training samples with at least one LF vote. We highlight the best result in **blue** and the second best result in **bold**.

| Dataset | MV | DawidSkene | MeTaL | FS | Snorkel | WSGAN-Vector | WSGAN-Encoder |
|---|---|---|---|---|---|---|---|
| AwA2 - A | 0.631 | 0.607 | 0.632 | 0.615 | 0.641 | **0.647** | **0.681** |
| AwA2 - B | 0.623 | 0.548 | 0.582 | 0.602 | 0.605 | **0.645** | **0.699** |
| DomainNet | 0.614 | **0.658** | 0.487 | 0.635 | 0.499 | **0.661** | 0.643 |
| MNIST | 0.775 | 0.729 | 0.766 | 0.773 | 0.766 | **0.782** | **0.813** |
| FashionMNIST | **0.735** | 0.717 | 0.730 | 0.734 | 0.729 | **0.737** | **0.744** |
| GTSRB | **0.816** | 0.619 | **0.815** | 0.671 | **0.814** | **0.815** | **0.823** |
| CIFAR10-A | 0.827 | **0.850** | 0.806 | 0.800 | 0.807 | **0.850** | **0.874** |
| CIFAR10-B | 0.716 | 0.677 | 0.708 | 0.708 | 0.707 | **0.725** | **0.731** |

**Evaluation Metrics** As common in related work, label model performance is compared based on the pseudolabel accuracy the models achieve on the training data, since programmatic weak supervision operates in a transductive setting. Weighted F1 and mean Average Precision are provided in Appendix E. To compare the quality of generated images, we use the Fréchet Inception Distance (FID) on color images, which has been shown to be consistent with human judgments (Heusel et al., 2017) and is used to measure performance of current state-of-the-art GAN approaches (Karras et al., 2021).To show the improvement in alignment of the auxiliary model $Q$'s predictions of the discrete latent code $b$ with the latent labels $y$, we track the Adjusted Rand Index (ARI) between the two.

## 4.2 RESULTS

We first discuss results comparing label model and image generation performance, before presenting the use of WSGAN for augmentation of the downstream classifier with synthetic samples. We repeat each experiment at least three times and average the results in our tables.

### 4.2.1 LABEL MODEL AND IMAGE GENERATION

**Label Model Performance** Table 2 shows a comparison of label model performance based on the accuracy of the posterior on the training data, without the use of any labeled data or validation sets. WSGAN-encoder largely outperforms alternative label models, while the simpler WSGAN-vector model performs competitively as well. These results hold according to additional metrics provided in Appendix C . Results with standard deviations over 5 random runs are provided in the Appendix in Table 11, indicating that many differences are significant.

**Discrete Latent Code Comparison** Figure 3 shows the evolving ARI between the ground truth and the auxiliary model $Q$'s prediction of the latent code on real data during model training. The figures show a large improvement in $Q$'s ability to uncover the unobserved class label structure when comparing WSGAN to InfoGAN, which is expected as WSGAN can take advantage of the weak signals encoded in LFs, while InfoGAN is completely unsupervised.

**Image Generation Performance** Table 3 compares FID of generated color images, suggesting that WSGAN models do take advantage of the weak supervision signal to improve $Q$, thereby improving the quality of generated images compared to an Info-GAN using the same base DCGAN architecture. WSGAN has a lower FID only on the GTSRB dataset, likely due to GTSRB's class imbalance and the dataset difficulty (43 classes, <23k samples).

Table 3: Color image generation quality (mean FID). The best scores are highlighted in **blue**.

| Dataset | InfoGAN | WSGAN-V | WSGAN-E |
|---|---|---|---|
| AwA2 - A | 41.62 | 36.74 | **34.71** |
| AwA2 - B | 41.62 | 36.79 | **34.52** |
| DomainNet | 53.98 | 50.16 | **44.35** |
| GTSRB | **69.67** | 75.27 | 73.96 |
| CIFAR10-A | 28.93 | 25.70 | **22.71** |
| CIFAR10-B | 33.50 | 26.17 | **24.41** |

### 4.2.2 DATA AUGMENTATION

We record the change in test accuracy for a ResNet-18 (He et al., 2016) end model when adding 1,000 synthetic WSGAN-encoder images $\tilde{x}$ to augment each dataset. While the increases are modest, the process is beneficial and does not require additional human labeling or data collection efforts. Adding larger amounts of synthetic samples did not lead to further increases, possibly due to the limited image quality achieved by the basic DCGAN design explored in this section.

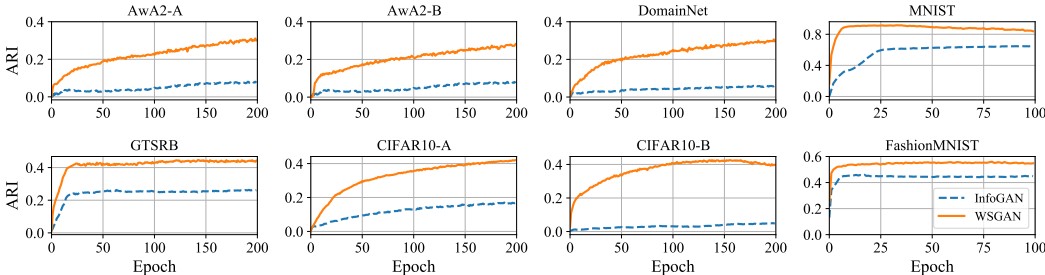

Figure 3: Adjusted Rand Index of the unobserved $y$ and the code predictions $Q(x)$ on real images $x$. Weak supervision allows WSGAN to better uncover latent $y$ compared to an unsupervised InfoGAN.

**Synthetic Images with Labeling Function Votes** The last column in Table 4 displays test accuracy increases by applying LFs $\lambda$ to synthetic images $\tilde{x}$. We obtain pseudolabels via $L_{A(\tilde{x})}(\lambda(\tilde{x}))$. We observe a modest average increase of 1.38%.

**Synthetic Images with Synthetic Pseudolabels** We can create pseudolabels with $F_1(Q(\tilde{x}))$, e.g. when LFs cannot be applied to synthetic images. With this, the second column of Table 4 shows an average increase in test accuracy of 1%, and up to 2.4%. We do not observe larger increases in accuracy by adding more generated images. Figure 10 shows a small number of generated images along with synthetic pseudolabel estimates. While $F_1(Q(x))$ could conceivably be used as a downstream classifier, the choices of network architecture are then constrained as it shares convolutional layers with $D$.

Table 4: Test accuracy increase when augmenting downstream model training with 1,000 synthetic images and pseudolabels (PLs). Synthetic PLs are obtained via $F_1(Q(\tilde{x}))$, LF PLs via $L_{A(\tilde{x})}(\lambda(\tilde{x}))$.

| Dataset | Synthetic PLs | LF PLs |
|---|---|---|
| AwA2 - A | 0.88% | 0.79% |
| AwA2 - B | 2.40% | 3.90% |
| DomainNet | 2.31% | 1.50% |
| MNIST | 1.60% | 1.71% |
| FashionMNIST | 0.29% | 0.34% |
| GTSRB | 0.40% | 0.02% |
| CIFAR10-A | 0.04% | - |
| CIFAR10-B | 0.30% | - |

**Synthetic Data Quality Checks** In addition to visually inspecting some generated samples and checking if conditionally generated samples reflect the target labels, we recommend checking the class balance in the pseudolabels of synthetic images before adding them to a downstream training set, as mode collapse in a trained GAN can potentially be diagnosed this way.

### 4.2.3 NETWORK ABLATION — STYLEWSGAN

We apply StyleWSGAN to weakly supervised LSUN scene categories (Yu et al., 2015), and to our CIFAR10-B dataset, please see Appendix C.2 for details. The results demonstrate WSGAN's complementarity with other GAN architectures and that it scales to images of higher resolution. On weakly supervised LSUN scene category images with a resolution of 256 by 256 pixels, StyleWSGAN achieves a mean FID of 7.54 (samples visualized in Fig. 8), while an unconditional, tuned StyleGAN2-ADA (Karras et al., 2020) achieves an FID of 8.41. On CIFAR10-B, StyleWSGAN achieves a mean FID of 3.79 (see generated images in Fig. 9), while also attaining a high label model accuracy of 0.736 (compare with Table 2). The unsupervised StyleGAN2-ADA, with the optimal, tuned settings identified in Karras et al. (2020), achieves an average FID of 3.85 on this subset. An unsupervised StyleInfoGAN that we created achieved a mean FID of 4.13.

## 5 RELATED WORK

**Programmatic Weak Supervision** Data programming (DP) (Ratner et al., 2016) is a popular weak supervision framework in which subject matter experts programmatically label data through multiple noisy sources of labels known as labeling functions (LFs). These LFs capture partial knowledge about an unobserved ground truth variable at better than random accuracy. In DP, a label model combines LF votes to provide an estimate of the unobserved ground truth, which is then used to train an end model using a noise-aware loss function. DP has been successfully applied to various domains including medicine (Fries et al., 2019; Dunnmon et al., 2020; Eyuboglu et al., 2021) and industry applications (Ré et al., 2020; Bach et al., 2019). Many works offer DP label models with improved properties, e.g., extensions to multitask models (Ratner et al., 2019), better computational efficiency (Fu et al., 2020), exploiting small amounts of labels as in semi-supervised

learning settings (Chen et al., 2021; Mazzetto et al., 2021a;b), end-to-end training (Cachay et al., 2021), interactive learning (Boecking et al., 2021), or extensions to structured prediction settings (Shin et al., 2022). See Zhang et al. (2022) for a more detailed survey.

**Programmatic Weak Supervision and Images** Our main focus is on applications of weak supervision to image data. On images, imperfect labels are often obtained from domain specific primitives and rules (Varma & Ré, 2018; Fries et al., 2019), rules defined on top of annotations by surrogate models (Varma & Ré, 2018; Chen et al., 2019; Hooper et al., 2021), rules defined on meta-data (Li & Fei-Fei, 2010; Chen & Gupta, 2015; Izadinia et al., 2015; Denton et al., 2015) or rules applied to a second paired modality such as text (Joulin et al., 2016; Wang et al., 2017; Irvin et al., 2019; Boecking et al., 2021; Dunnmon et al., 2020; Saab et al., 2019; Eyuboglu et al., 2021).

**Generative Models and Disentangled Representations** Among the numerous existing approaches to generative modeling, in this work we focus on generative adversarial networks (GANs) (Goodfellow et al., 2014). We are particularly interested in work that aims to learn disentangled representations (Chen et al., 2016; Lin et al., 2020) that can align with class variables of interest. Chen et al. (2016) introduce InfoGAN, which learns interpretable latent codes. This is achieved by maximizing the mutual information between a fixed small subset of the GAN's input variables and the generated observations. Gabbay & Hoshen (2020) present a unified formulation for class and content disentanglement as well as a new approach for class-supervised content disentanglement. Nie et al. (2020) study semi-supervised high-resolution disentanglement learning for the state-of-the-art StyleGAN architecture. A potential downside to modeling latent factors in generative models is a decrease in image quality of generated samples that has been noted when disentanglement terms are added (Burgess et al., 2018; Khrulkov et al., 2021).
Prior work has studied how to integrate additional information into GAN training, in particular ground truth class labels (Mirza & Osindero, 2014; Salimans et al., 2016; Odena, 2016; Odena et al., 2017; Brock et al., 2019; Thekumparampil et al., 2018; Miyato & Koyama, 2018; Lučić et al., 2019), also considering noisy scenarios (Kaneko et al., 2019). However, in the programmatic weak supervision setting, having multiple noisy sources of imperfect labels that include abstains present large hurdles to similar conditional modeling. Some prior work uses other weak formats of supervision to aid specific aspects of generative modeling. For example, Chen & Batmanghelich (2020) propose learning disentangled representation using user-provided ground-truth pairs. Yet, prior work does not fuse programmatic weak supervision frameworks and generative models, and so are limited to one-off techniques to solely improve generative models.

**Using GANs for Data Augmentation** An exciting application of GANs is to generate additional samples for supervised model training. The challenge is to produce sufficiently high-quality samples. For example, Abbas et al. (2021) use a conditional GAN to generate synthetic images of tomato plant leaves for a disease detection task. GANs for data augmentation are also popular in medical imaging (Yi et al., 2019; Motamed et al., 2021; Hu et al., 2019). For example, Hu et al. (2019) use an InfoGAN-like model to learn cell-level representations in histopathology, Motamed et al. (2021) augment radiology data, and Pascual et al. (2019) generate synthetic epileptic brain activities.

## 6 CONCLUSION

We studied the question of how to build an interface between two powerful techniques that operate in the absence of labeled data: generative modeling and programmatic weak supervision. Our fusion of the two, a weakly supervised GAN (WSGAN), defines an interface that aligns structures discovered in its constituent models. This leads to three improvements: first, better quality pseudolabels compared to weak supervision alone, boosting downstream performance. Second, improvement in the quality of the generative model samples. Third, it enables data augmentation via generated samples and pseudolabels, further improving downstream model performance without additional burden on users.

Standard failure cases of GANs such as mode collapse still apply to the proposed approach. However, we do not observe that WSGAN is more susceptible to such failures than the approaches we compare to. For future work, we are interested in other modalities, exploiting for instance generative models for graphs and time series. Further, motivated by the performance of WSGAN, we seek to extend the underlying notion of interfaces between models to a variety of other pairs of learning paradigms. Limitations of the proposed approach include common GAN restrictions such as the types of data that can be modeled and the number of unlabeled samples required to fit distributions, and also known difficulties of acquiring weak supervision sources of sufficient quality for image data.

ACKNOWLEDGMENTS

This work was partially supported by a Space Technology Research Institutes grant from NASA's Space Technology Research Grants Program and the Defense Advanced Research Projects Agency's award FA8750-17-2-0130. This work was also supported in part by NSF (#1651565, #CCF2106707) and ARO (W911NF2110125), and the Wisconsin Alumni Research Foundation (WARF).

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

APPENDIX

---

**Algorithm 1** Pseudocode for the proposed WSGAN loss term which is added to the basic InfoGAN loss. Input images and LFs are assumed to be filtered to only contain samples with at least 1 non-abstaining LF vote.

---

   **input:** Batch of real images $X$ and one-hot encoded LFs $\Lambda$, label model $L$, networks $Q, A, F_1, F_2$, WSGAN mode $m$, number of classes $C$, current epoch $i$, $\gamma$ decay parameter.
      $\hat{b}, Z = Q(X)$ # get predicted code and image features $Z$
      **if** $m ==$ *"vector"*:
         $\theta = A_\theta()$ # WSGAN-vector: get weight vector.
      **else:**
         $\theta = A(Z.detach())$ # WSGAN-encoder: predict weights using image features $Z$.
      $\hat{y} = L(\Lambda, \theta)$ # Get label estimate from labelmodel using weights $\theta$.
      # Compute cross-entropy losses
      $loss = celoss(F_1(\hat{b}), \hat{y}.\text{detach}())$
      $loss \mathrel{+}= celoss(F_2(\hat{y}), \hat{b}.\text{detach}())$
      $loss \mathrel{+}= C/(i \times \gamma + 1)mse(\theta, \vec{1} \times 0.5)$ # add decaying loss keeping weights uniform.
   **return** $loss$

---

## A   Implementation Details and Complexity

Code for WSGAN can be found at `https://github.com/benbo/WSGAN-paper`.

**(WS)GAN Models**   The following design choices were used for the experiments conducted with simple DCGAN base networks (as opposed to the settings used in the StyleGAN ablations).

*Generator G, Discriminator D, and auxilliary model Q*: Figures 5 and 4 show the simple DC-GAN (Radford et al., 2015) based generator and discriminator architectures we use in WSGAN for experiments with $32 \times 32$ images. As mentioned in the main paper, $Q$ and $D$ are neural networks that generally share all convolutional layers, with a final fully connected layer to output predictions. We follow the same structure in our experiments. We set the dimension of the noise variable $z$ to 100, and of $b$ equal to the number of classes. We sample $z$ from a normal distribution and $b$ from a uniform discrete distribution.

*Accuracy Encoder A*: For WSGAN-Vector, $A$ is simply a parameter vector of the same length as the number of labeling functions. For WSGAN-Encoder, we use image features obtained from the shared convolutional layers of $Q$ and $D$, which we detach from the computational graph before passing them on to an MLP prediction head. For images with $32 \times 32$ pixels, the feature vector obtained from the shared convolutional layers is of size $512 * 16$. The MLP head of $A$ is set to have three hidden layers of size $(256, 128, 64)$, with ReLU activations, and an output layer the size of the number of labeling functions followed by a sigmoid function. We did not observe significant changes in performance when we change the MLP to be shallower or wider. However, for large numbers of LFs, one should consider increasing the width the MLP.

*Mappings $F1, F2$*: We set $F1$ and $F2$ to each be simple linear models with a softmax at the output, and set the input and output size of each to the number of classes.

**(WS)GAN Training**   We use the same hyperparameter settings for all datasets. We train all GANs for a maximum of 200 epochs. We use a batch size of 16 and find that a lower batch size leads to more frequent convergence of the generator and discriminator. We also conducted ablation experiments with a batch size of 8 and 32 and found no significant difference in FID image generation quality or label model accuracy. For WSGAN, we use four optimizers, one for each of the different loss terms: discriminator training, generator training, the Info loss term, and the WSGAN loss term. We use Adam for all optimizers and set the learning rates as follows: $4 \times 10^{-4}$ for $D$, $1 \times 10^{-4}$ for $G$, $1 \times 10^{-4}$ for the info loss term, and $8 \times 10^{-5}$ for the WSGAN loss term. We follow the same settings for InfoGAN training for the components shared with WSGAN.

**(WS)GAN Training and Failure Cases**   While WSGAN is still susceptible to the common GAN failure cases of its base networks, such as mode collapse, we empirically find WSGAN training to be more stable than training a GAN that also learns a discrete latent code but uses no weak supervision signals (InfoGAN), despite the high level of noise in our weak supervision sources. InfoGAN failed to converge more frequently.

To help train the DCGAN networks successfully, we find that employing discriminator label flipping (randomly calling a tiny percentage of real samples fake and vice versa) and label smoothing (adding small amounts of noise to the real target of 1.0 and fake target of 0.0) stabilizes and improves GAN training. Despite employing these tricks, we were unable to avoid occasional convergence failures. Fortunately, monitoring the generator and discriminator losses, inspecting the quality of generated images, or tracking image quality metrics such as FID allows one to easily discard failed runs or to pick model checkpoints from earlier iterations before a failure, without requiring labeled data.

**StyleWSGAN Model Setup and Training**   We adapt StyleGAN2-ADA (Karras et al., 2020) to build a StyleWSGAN Model as well as a StyleInfoGAN. The generator architecture follows is the same approach as a class-conditional StyleGAN generator: the sampled code is embedded to a $d$-dimensional vector via a linear layer and then concatenated with the original latent code, after each is normalized. This concatenated vector is then passed to the StyleGAN mapping network. We find the relationship between the number of layers of the StyleGAN mapping network and the size of the embedded sampled code $d$ to be crucial for StyleWSGAN. When the mapping network is too shallow, as in the tuned CIFAR10 settings in (Karras et al., 2020), a large $d$ can lead to training instability for StyleWSGAN and StyleInfoGAN.

We use separate optimizer settings for each loss term, and set the learning rate for the Info term (added term of Equation 2) and the WSGAN term (added term of Equation 4 plus decay penalty) to a factor of $2/10$ of the base learning rate in StyleGAN. This results in a learning rate of 0.0005 for the added WSGAN terms in our experiments, while we maintain a learning rate of 0.0025 for the original StyelGAN terms. Due to the use of different learning rates in the separate optimizers, the added loss terms are not scaled, and the hyper-parameters $\alpha, \beta$ are set to 1.

For the CIFAR10 experiments we largely follow the settings used in (Karras et al., 2020): no style mixing, no path length regularization, no ResNet D. However, we increase the depth of the mapping network from 2 to 6, we decrease the size of the code embedding to 200, and continue training until the discriminator has seen a total of 50M real images. A mapping network of depth 4, and a code embedding size of 50 also lead to good performance, performing only slightly worse measured by both FID and label model accuracy.

For the LSUN experiments, we train StyleWSGAN until the discriminator has seen a total of 35M real images, and the baseline StyleGAN2-ADA that we compare to is trained until the discriminator has seen a total of 50M real images. We largely follow the settings used for 256 x 256 images in (Karras et al., 2020), but disable style mixing and path length regularization. We set the size of the discrete code embedding to 50.

**End Model Training**   For all datasets, we train a ResNet-18 (He et al., 2016) for 100 epochs, using Adam and a learning rate scheduler. The learning rate scheduler uses a small validation set to make adjustments to the learning rate.

**Image Augmentation**   We use the following random image augmentation functions during DC-GAN and endmodel training for color images: random crop and resize (cropping out a maximum height/width of 13%), random sharpness adjustment ($p = 0.2$), random Gaussian blur ($p = 0.1$), and random color jitter.

**Label Models**   To compare to related work, we use implementations of label models made available via WRENCH (Zhang et al., 2021).

**Complexity**   WSGAN shares the same operations as InfoGAN and adds some additional steps on real samples that have at least one LF vote, which slightly increases the required computation. Recall that $C$ denotes the number of classes, $m$ the number of LFs, and $x$ an image of a real sample. Further, let $n_w$ denote the number of samples that have at least one weak label vote from any LF, let

```
============================================================================
Layer (type:depth-idx)                  Output Shape            Param #
============================================================================
InfoDCDiscriminator                     --                      --
├─Sequential: 1-1                       [16, 512, 4, 4]         --
│    └─Conv2d: 2-1                       [16, 64, 32, 32]        1,792
│    └─LeakyReLU: 2-2                    [16, 64, 32, 32]        --
│    └─Conv2d: 2-3                       [16, 64, 16, 16]        65,600
│    └─LeakyReLU: 2-4                    [16, 64, 16, 16]        --
│    └─Conv2d: 2-5                       [16, 128, 16, 16]       73,856
│    └─LeakyReLU: 2-6                    [16, 128, 16, 16]       --
│    └─Conv2d: 2-7                       [16, 128, 8, 8]         262,272
│    └─LeakyReLU: 2-8                    [16, 128, 8, 8]         --
│    └─Conv2d: 2-9                       [16, 256, 8, 8]         295,168
│    └─LeakyReLU: 2-10                   [16, 256, 8, 8]         --
│    └─Conv2d: 2-11                      [16, 256, 4, 4]         1,048,832
│    └─LeakyReLU: 2-12                   [16, 256, 4, 4]         --
│    └─Conv2d: 2-13                      [16, 512, 4, 4]         1,180,160
│    └─LeakyReLU: 2-14                   [16, 512, 4, 4]         --
│    └─Dropout: 2-15                     [16, 512, 4, 4]         --
├─Sequential: 1-2                       [16, 1, 1, 1]           --
│    └─Conv2d: 2-16                      [16, 1, 1, 1]           8,192
============================================================================
```

Figure 4: The DCGAN discriminator architecture used in our experiments with 32 x 32 images.

```
============================================================================
Layer (type:depth-idx)                  Output Shape            Param #
============================================================================
DCGeneratorThree                        --                      --
├─Linear: 1-1                           --                      2
├─Sequential: 1-2                       [16, 3, 32, 32]         --
│    └─ConvTranspose2d: 2-1              [16, 512, 4, 4]         901,632
│    └─BatchNorm2d: 2-2                  [16, 512, 4, 4]         1,024
│    └─ReLU: 2-3                         [16, 512, 4, 4]         --
│    └─ConvTranspose2d: 2-4              [16, 256, 8, 8]         2,097,408
│    └─BatchNorm2d: 2-5                  [16, 256, 8, 8]         512
│    └─ReLU: 2-6                         [16, 256, 8, 8]         --
│    └─ConvTranspose2d: 2-7              [16, 128, 16, 16]       524,416
│    └─BatchNorm2d: 2-8                  [16, 128, 16, 16]       256
│    └─ReLU: 2-9                         [16, 128, 16, 16]       --
│    └─ConvTranspose2d: 2-10             [16, 64, 32, 32]        131,136
│    └─BatchNorm2d: 2-11                 [16, 64, 32, 32]        128
│    └─ReLU: 2-12                        [16, 64, 32, 32]        --
│    └─ConvTranspose2d: 2-13             [16, 3, 32, 32]         1,731
│    └─Tanh: 2-14                        [16, 3, 32, 32]         --
============================================================================
```

Figure 5: The DCGAN generator architecture used in our experiments with 32x32 images.

$q$ denote the number of steps required for a forward pass through $Q$ to obtain image features and the discrete code prediction, and $a$ denote the number of steps for a forward pass through the MLP $A$. For a forward pass, WSGAN increases the complexity compared to InfoGAN in each epoch by $\Theta(n_w(a + q + m + 2C^2 + C(m + 8)))$. Note that $q$ may be eliminated for the forward pass through careful implementation as the image features are already obtained for the basic InfoGAN update. In practice, in our experiments the computational overhead, including for additional data loading of the LFs, leads to a modest increase in runtime (measured in bps, denoting batches per second) of the weakly supervised WSGAN over the unsupervised InfoGAN, as follows. InfoGAN: 14bps, WSGAN Encoder: 7.8bps, WSGAN Vector: 8.6bps (NVIDIA RTX A6000, batch size 16). In terms of parameters, WSGAN shares the same generator G and discriminator components D,Q as InfoGAN, and adds additional label model parameters. The overall number of parameters in our experiments with 32 x 32 images are: InfoGAN 6.7M, WSGAN 8.8M.

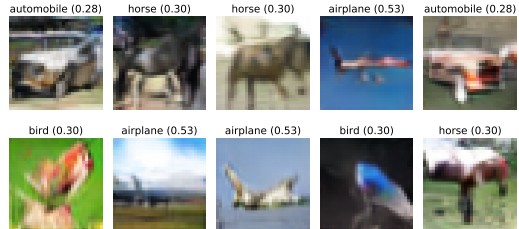

Figure 6: Some synthetic images and pseudolabels generated by the proposed WSGAN with a DCGAN base-architecture, learned from weakly supervised CIFAR10. We note that WSGAN is able to generate images and estimate their labels, even for images where no weak supervision sources provide information (see end of Section 3.1 for details).

# B    DATASET DETAILS

- *CIFAR10* contains 32x32 color images of 10 different classes. We create two different subsets of CIFAR10. One set (used for experiments CIFAR10-C,D) uses the full training set of CIFAR10 (minus 300 samples held out for downstream validation), while the second (used for experiments CIFAR10-A,B,E,F) is a random subset of 30,000 training images.

- *MNIST* and *FashionMNIST* both contain 28x28 grayscale images, which we resize to 32x32. For both, we use a random sample of 30,000 images from the training data for our experiments. SSL-based labeling functions are fine-tuned on small, random subsets of the remaining training data of each dataset.

- *GTSRB* contains 64x64 color images of German traffic signs. We use 22,640 random images from the full training dataset during our experiments, while random subsets of the remaining images in the original training data are used to finetune the SSL-based labeling functions.

- The original *DomainNet* (Peng et al., 2019) dataset contains 345 classes of images in 6 different domains [1]. As our dataset, following Mazzetto et al. (2021a) we use the images in the real domain and select the 10 classes with the largest number of instances in this domain. Because of the small size of the resulting dataset, we resize the images to 32 x 32 in our experiments.

- *Animals with Attributes 2* (AwA2) (Mazzetto et al., 2021a) is an image dataset with known general attributes for each class, divided into 40 seen and 10 unseen classes. Because of the small size of the resulting dataset once LFs are created, we resize the images to 32 x 32 in our experiments.

- *LSUN scene categories* see Section C.2 for details.

## B.1    LABELING FUNCTION DETAILS

- *Synthetic*: based on the true class label, we create synthetic, unipolar LFs via the following procedure: for each LF, we sample a class label, an error rate, and a propensity (i.e., the percentage of samples where the LF casts a vote, also referred to as coverage). Given the target label, we then sample true positives and false positives at random to achieve the desired LF accuracy and propensity.

- *Domain transfer*: these LFs are used in our DomainNet dataset experiments. We follow Mazzetto et al. (2021a), and derive weak supervision sources for a multiclass classification task of the real images contained in the DomainNet (Peng et al., 2019) dataset. First, we set our target domain to real images and select the 10 classes with the largest number of instances in this domain. As LFs, we then train classifiers using the selected classes within the remaining five domains, and apply these trained classifiers to the unseen images in the target domain of real images to obtain weak labels.

- *Attribute heuristics*: we create two sets of LFs for the Animals with Attributes 2 (AwA2) (Mazzetto et al., 2021a) image classification dataset. Following Mazzetto et al.

---

[1]Real, painting, sketch, clipart, infograph, quickdraw.

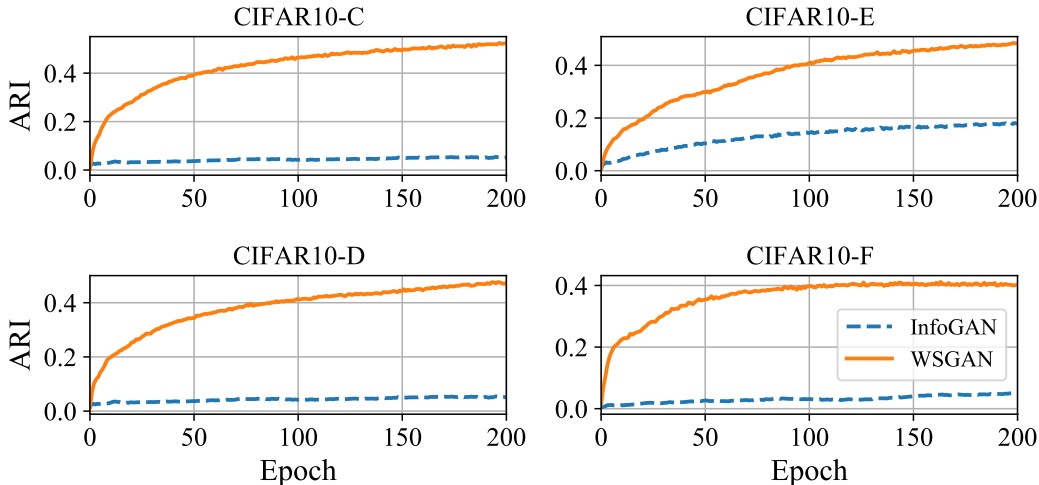

Figure 7: We here show the Adjusted Rand Index (ARI) for the additional CIFAR experiments. The plots show the ARI between the unobserved class label $y$ and the discrete code prediction by the auxiliary model $Q(x)$ on real image $x$, during training. Weak supervision allows WSGAN to better uncover the latent class structure compared to an unsupervised InfoGAN.

(2021b;a), we train one-vs-rest attribute classifiers using the 40 seen classes of the AwA2 dataset. These classifiers are applied to the 10 unseen classes to produce weak attribute labels. At this stage, we discard attribute classifiers which perform worse than random. We create an 85%/5%/10% train/validation/test split of the 10 unseen classes which we use to define decision trees to produce weak labels on the bases of weak attribute predictions. We create the 29 unipolar LFs for *AwA2-A* by training 3 one-vs-rest decision trees per each of the 10 classes on 100 random samples from the training set. To create a slightly easier set, we create the 32 unipolar LFs used in *AwA2-B* by training 80 decision trees, retaining one random tree specializing in each class, and then selecting all remaining ones where validation accuracy is higher than $0.65$.

- *SSL-based*: The base representations are learned on unsupervised ImageNET with Sim-CLR (Chen et al., 2020). The trained network is used to obtain features for our image datasets. We then train shallow MLP networks on a few hundred held-out samples to predict a randomly sampled target label at a randomly sampled target accuracy. The accuracy is validated to be within range of the target accuracy on another small amount of held-out data. Thus, during their creation these unipolar LFs are never trained or evaluated on the WSGAN training data or the downstream test data.

## C   ADDITIONAL EXPERIMENTS

### C.1   WSGAN WITH A DCGAN BASE-ARCHITECTURE

Table 5: Additional datasets and labeling function (LF) characteristics used to evaluate the proposed WSGAN model. Acc denotes accuracy, while Coverage denotes the number of samples where the LF does not abstain.

| Dataset | #Classes | #LFs | #Samples | Mean LF Acc | Min LF Acc | Max LF Acc | Mean Coverage | LF Type |
|---------|----------|------|----------|-------------|------------|------------|---------------|---------|
| CIFAR10-C | 10 | 20 | 49,700 | 0.747 | 0.621 | 0.879 | 0.048 | Synthetic |
| CIFAR10-D | 10 | 40 | 49,700 | 0.760 | 0.621 | 0.898 | 0.052 | Synthetic |
| CIFAR10-E | 10 | 40 | 30,000 | 0.761 | 0.624 | 0.896 | 0.056 | Synthetic |
| CIFAR10-F | 10 | 40 | 30,000 | 0.728 | 0.531 | 0.912 | 0.046 | SSL, finetuning |

For further evaluation of WSGAN with a DCGAN base architecture, we created additional weakly supervised image datasets based on CIFAR10 by varying the number of samples and the type

Table 6: Additional datasets to evaluate WSGAN with a DCGAN base architecture. This table shows average posterior accuracy of various label models on training samples with at least one LF vote. We highlight the best result in **blue** and the second best result in **bold**.

| Dataset | MV | DawidSkene | MeTaL | FS | Snorkel | WSGAN-Vector | WSGAN-Encoder |
|---------|------|------------|-------|-------|---------|--------------|---------------|
| CIFAR10-C | 0.762 | **0.778** | 0.751 | 0.764 | 0.757 | **0.778** | **0.796** |
| CIFAR10-D | 0.831 | **0.861** | 0.819 | 0.805 | 0.812 | 0.854 | **0.865** |
| CIFAR10-E | 0.865 | **0.902** | 0.845 | 0.827 | 0.849 | 0.898 | **0.917** |
| CIFAR10-F | 0.687 | 0.601 | 0.682 | 0.677 | 0.678 | **0.691** | **0.702** |

Table 7: Additional datasets: color image generation quality measured by average Fréchet Inception Distance (FID). The best scores for each dataset are highlighted in **blue**.

| Dataset | InfoGAN | WSGAN-V | WSGAN-E |
|---------|---------|---------|---------|
| CIFAR10-C | 33.64 | **24.11** | 26.00 |
| CIFAR10-D | 33.64 | 24.09 | **23.78** |
| CIFAR10-E | 28.93 | **21.97** | 22.63 |
| CIFAR10-F | 33.50 | 24.59 | **22.54** |

of labeling function, see CIFAR10 dataset details in Table 5. The proposed WSGAN approach outperforms related approaches in these experiments as well. The label model accuracy results are shown in Table 6, while additional metrics including F1 are shown in Section E. Image generation quality results are provided in Table 7. Finally, a comparison between the latent discrete variable of WSGAN and InfoGAN is given in Figure 7, which shows how the Adjusted Rand Index evolves between the unobserved class labels and the latent discrete variable modeled by auxiliary model $Q$.

## C.2 STYLEWSGAN

Please see Section A for implementation details and hyperparameter settings in our StyleGAN experiments. Dataset statistics for this section are shown in Table 8.

**LSUN scene categories**   To test the proposed WSGAN on higher resolution images with a Style-GAN base architecture, we create a balanced subset of the LSUN scene categories dataset (Yu et al., 2015). The dataset contains 10 classes (i.e. 10 different scene categories) and we center-crop and resize images to 256 by 256 pixels. We sample an equal number of images from each of the 10 classes for a final dataset size of 1,212,270 images. As weak supervision sources, we create 30 SSL-based LFs by training classifiers on small amounts of held-out data using image features learned via self-supervised learning, as described in Section B.1.

StyleWSGAN achieves an average FID of 7.54 on this dataset. An unconditional StyleGAN2-ADA achieves an FID of 10.3 with the settings for 256 by 256 images set in (Karras et al., 2020), and an FID of 8.41 when we turn off path length regularization and style mixing. Note that unconditional StyleGAN results on LSUN images with lower FID scores reported in related work are generally obtained by training on a single LSUN scene or object category, rather than on multiple categories simultaneously, as in our experiments, which results in a more challenging setup.

The average WSGAN labelmodel accuracy on this weakly supervised LSUN dataset is 0.766. Other label models obtain the following average accuracies: DawidSkene 0.765, Majority Vote 0.76, FlyingSquid 0.739, Snorkel MeTaL 0.740, and Snorkel 0.728.

**CIFAR10**   First, Figure 9 shows synthetic images by StyleWSGAN on the weakly supervised CIFAR10-B dataset, which uses SSL-based LFs. These LFs are quite noisy, with a mean LF accuracy of 0.736, which is reflected in the noisy class-conditional samples that can be inspected in Figure 9. On this dataset, StyleWSGAN achieves a mean FID of 3.79, while also attaining a high label model accuracy of 0.736. The unsupervised StyleGAN2-ADA, with the optimal, tuned settings identified in (Karras et al., 2020), achieves an average FID of 3.85 on this subset.

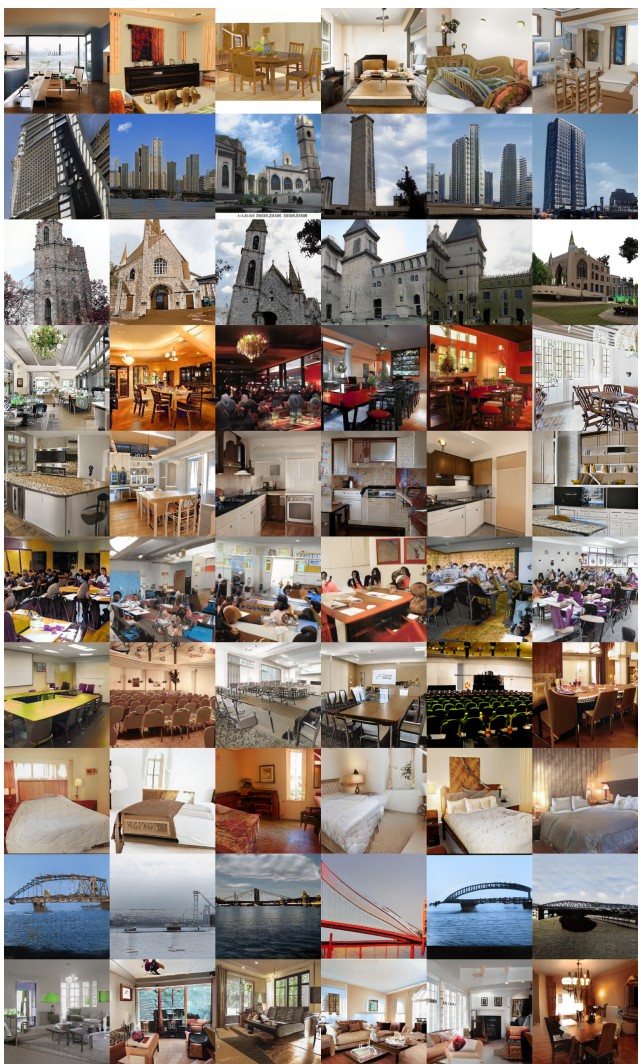

Figure 8: Synthetic images learned by StyleWSGAN on a weakly supervised subset of the LSUN scene category dataset.

We create an additional weakly supervised CIFAR10 with lower noise LFs, to see if such a setting can lead to results that are better than the state-of-the-art (SOTA) unsupervised image generation quality on the full CIFAR10 dataset reported in (Karras et al., 2020). For this experiment, we create LFs by randomly introducing errors and abstains to the ground-truth vector. For the LFs, we set a minimum accuracy of 0.8 and a maximum accuracy of 0.95 and create 20 LFs. This dataset contains 48000 samples, has a mean LF accuracy of 0.888, and a mean coverage of 0.102 (meaning that an LF on average abstains on $\sim 89.8\%$ of the dataset). For this dataset, StyleWSGAN achieves an FID of 2.84, which is better than the SOTA unsupervised result reported in (Karras et al., 2019) of 2.92 FID on the full 50k CIFAR10 samples, but shy of the performance of the conditional StyleGAN (Karras et al., 2019) which uses projection discrimination and has access to all ground-truth labels and achieves and FID of 2.42.

Table 8: Additional datasets used to evaluate StyleWSGAN. Acc denotes accuracy, while Coverage denotes the number of samples where an LF does not abstain.

| Dataset | #Classes | #LFs | #Samples | Mean LF Acc | Min LF Acc | Max LF Acc | Mean Coverage | LF Type |
|---|---|---|---|---|---|---|---|---|
| CIFAR10 - low noise LFs | 10 | 20 | 48,000 | 0.888 | 0.816 | 0.949 | 0.102 | Synthetic |
| LSUN scene categories | 10 | 30 | 1,212,270 | 0.736 | 0.624 | 0.873 | 0.098 | SSL-based |

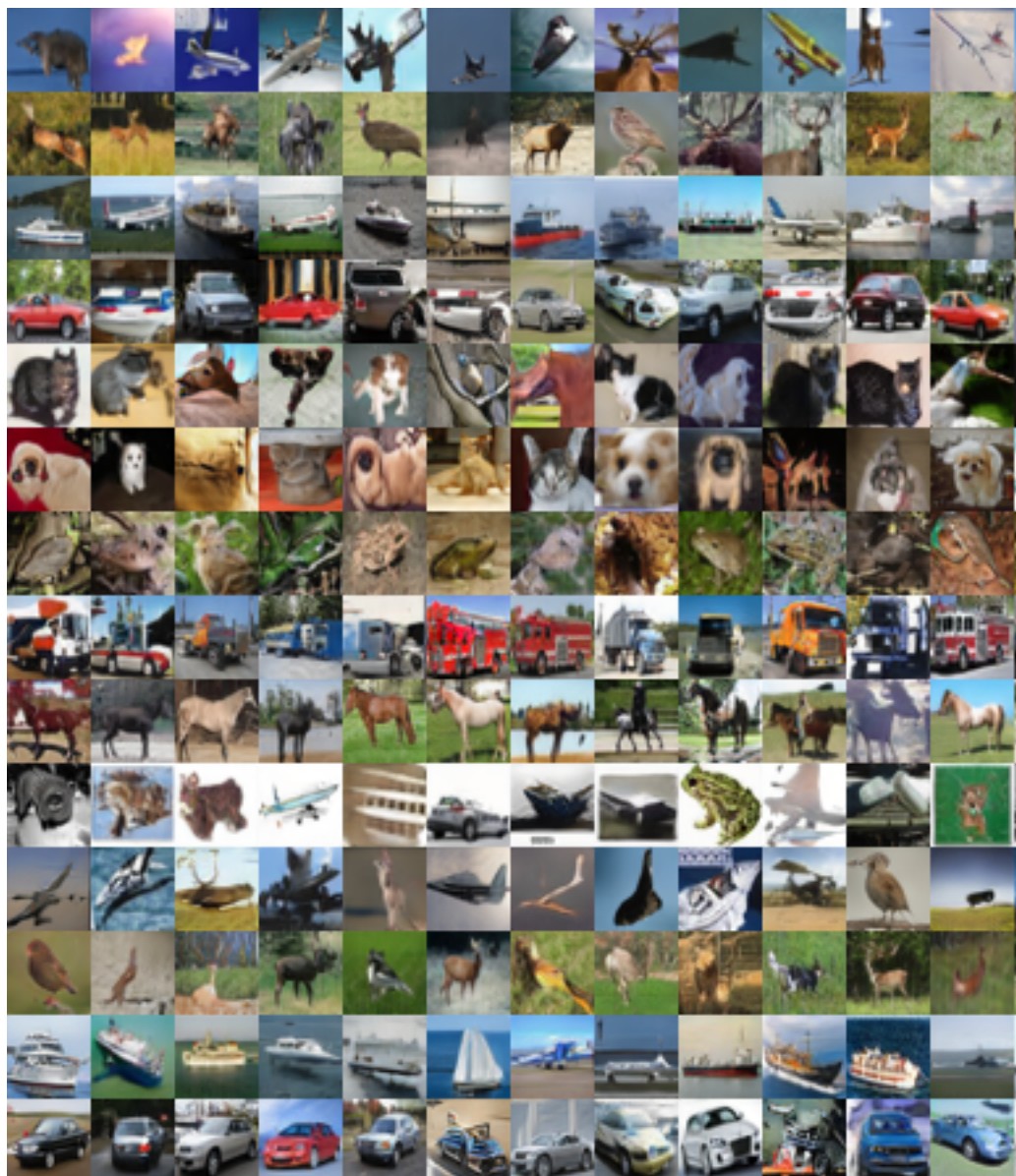

Figure 9: Synthetic images learned on the CIFAR10-B subset by StyleWSGAN, which a version of our WSGAN that built on StyleGAN2-ADA rather than a simple DCGAN as in our main experiments.

## D   ADDITIONAL BASELINES

We compare against two additional baselines. First, we train a generative model that is conditioned on pseudolabel information with the aim of improving image generation performance; we use pseudolabels provided by established weak-supervision label models in this role. Second, we use a basic generative model to produce synthetic samples that augment a downstream classifier (with weak labels provided by outputs of weak supervision sources applied to the synthetic images). These two baselines represent the straightforward way to use weak supervision to improve generative modeling (and vice-versa). We observe that such naive combinations struggle compared to our proposed approach, further motivating the importance of the interface in WSGAN.

Table 9: A comparison to using an ACGAN with pseudolabels. Image generation quality is measured by average Fréchet Inception Distance (FID). The best scores for each dataset are highlighted in **blue**.

| Dataset | InfoGAN | ACGAN | ACGAN (crisp) | WSGAN-V | WSGAN-E |
|---|---|---|---|---|---|
| AwA2 - A | 41.62 | 51.32 | 47.21 | 36.74 | **34.71** |
| AwA2 - B | 41.62 | 53.73 | 50.03 | 36.79 | **34.52** |
| DomainNet | 51.88 | 61.96 | 47.32 | 51.16 | **45.6** |
| CIFAR10-A | 28.93 | 79.15 | 25.53 | 25.7 | **22.71** |
| CIFAR10-C | 33.64 | 36.53 | 26.61 | **24.11** | 26.0 |
| CIFAR10-D | 33.64 | 36.81 | 45.1 | 24.09 | **23.78** |
| CIFAR10-E | 28.93 | 80.43 | 33.05 | **21.97** | 22.63 |

Table 10: Baseline comparisons using an InfoGAN to create synthetic images, applying LFs to the synthetic images, and then using established label models to synthesize the weak labels in to a pseudolabel resulting in weakly labeled fake images. The table shows the change in test accuracy by augmenting the downstream classifier training data with such 1,000 synthetic images and corresponding pseudo labels. Experiments are conducted on a subset of the datasets where labeling functions can be applied to synthetic images.

| Dataset | WSGAN | InfoGAN + Snorkel | InfoGAN + DawidSkene |
|---|---|---|---|
| AwA2 - A | 0.79% | -0.63% | -1.26% |
| AwA2 - B | 3.90% | -1.01% | -1.77% |
| DomainNet | 1.50% | 0.02% | -3.14% |

## D.1 CONDITIONAL IMAGE GENERATION WITH PSEUDOLABELS OR RAW WEAK SUPERVISION VOTES

As an additional GAN baseline to our proposed WSGAN, we slightly adapt an Auxiliary Classifier Generative Adversarial Network (ACGAN) (Odena et al., 2017) to condition it on pseudolabels. The ACGAN is run on all data, but the auxiliary loss on real data with pseudolabels is only used for samples where at least one labeling function does not abstain. We create two versions: (1) using probabilistic pseudolabels with a soft cross-entropy loss, and (2) using *hard/crisp* labels with a cross-entropy loss. To provide the strongest possible baseline in this experiment, we obtain the pseudolabels via the Dawid-Skene label model as it attains the best performance on average over all datasets compared to other related label models. Results are provided in Table 9, showing that this baseline approach is frequently unable to overcome the noise in the pseudolabels to improve over the InfoGAN results, and that it does not perform better than WSGAN with an encoder. Furthermore, it was much more difficult to train these models and they frequently failed to converge.

We also attempted to train different types of conditional GANs (ACGAN and a GAN with projection discrimination) conditioned on the raw weak supervision votes, but were unable to obtain reasonable performance as the models failed to converge.

## D.2 DATA AUGMENTATION FOR DOWNSTREAM CLASSIFICATION WITH SYNTHETIC IMAGES

In this experiment, we augment the training set for a downstream classifier with synthetic images. As baselines, we generate synthetic images $\tilde{x}$ with an InfoGAN, and then apply the image labeling functions $\lambda$ to the generated images to obtain LF votes $\lambda(\tilde{x})$. Pseudolabels for the synthetic images are then obtained by fitting label models to the real training data and then applying the label models to the labeling function outputs on the synthetic data. Table 10 compares InfoGAN + Snorkel and InfoGAN + DawidSkene baselines to the improvements in test accuracy obtained by using WSGAN and shows that we were unable to obtain improvements in downstream accuracy with the baselines, possibly due to performance of the InfoGAN on these small datasets.

Table 11: In this table, we include standard deviations for the posterior accuracy of various label models on training samples with at least one LF vote, computed over five random runs. Due to a limited computational budget, we were unable to accumulate five runs for all datasets and model combinations.

| Dataset | DawidSkene | MeTaL | FS | Snorkel | WSGAN-Encoder |
|---|---|---|---|---|---|
| AwA2 - A | 0.607(±0.029) | 0.632(±0.002) | 0.615(±0.003) | 0.641(±0.001) | **0.681**(±0.011) |
| DomainNet | **0.658**(±0.000) | 0.487(±0.004) | 0.635(±0.000) | 0.499(±0.015) | 0.643(±0.003) |
| MNIST | 0.729(±0.000) | 0.766(±0.001) | 0.773(±0.000) | 0.766(±0.001) | **0.813**(±0.004) |
| FashionMNIST | 0.717(±0.002) | 0.730(±0.001) | 0.734(±0.001) | 0.729(±0.001) | **0.744**(±0.002) |
| GTSRB | 0.619(±0.001) | **0.815**(±0.002) | 0.679(±0.001) | **0.814**(±0.000) | **0.823**(±0.001) |
| CIFAR10-A | **0.850**(±0.001) | 0.806(±0.001) | 0.800(±0.000) | 0.807(±0.002) | **0.874**(±0.002) |
| CIFAR10-B | 0.677(±0.000) | 0.708(±0.001) | 0.708(±0.000) | 0.707(±0.000) | **0.731**(±0.004) |

Table 12: Weighted mean average precision of various label models on training samples with at least one LF vote. We highlight the best result in **blue** and the second best result in **bold**.

| Dataset | MV | DawidSkene | MeTaL | FS | Snorkel | WSGAN-Vector | WSGAN-Encoder |
|---|---|---|---|---|---|---|---|
| AwA2 - A | 0.616 | 0.661 | 0.653 | 0.627 | 0.653 | **0.672** | **0.737** |
| AwA2 - B | 0.591 | 0.652 | 0.662 | 0.642 | 0.668 | **0.681** | **0.743** |
| DomainNet | 0.599 | **0.702** | 0.630 | 0.654 | 0.621 | 0.679 | **0.795** |
| MNIST | 0.684 | 0.772 | 0.784 | 0.765 | 0.785 | **0.792** | **0.870** |
| FashionMNIST | 0.620 | 0.712 | 0.691 | 0.686 | 0.692 | **0.703** | **0.742** |
| GTSRB | 0.718 | 0.731 | 0.761 | 0.714 | **0.772** | 0.768 | **0.808** |
| CIFAR10-A | 0.796 | 0.866 | 0.855 | 0.838 | 0.854 | **0.878** | **0.912** |
| CIFAR10-B | 0.594 | 0.659 | 0.658 | 0.631 | 0.666 | **0.678** | **0.732** |
| CIFAR10-C | 0.664 | 0.763 | 0.758 | 0.737 | 0.751 | **0.780** | **0.825** |
| CIFAR10-D | 0.788 | 0.896 | 0.889 | 0.876 | 0.878 | **0.901** | **0.908** |
| CIFAR10-E | 0.880 | **0.954** | 0.942 | 0.924 | 0.940 | 0.950 | **0.959** |
| CIFAR10-F | 0.561 | 0.658 | 0.66 | 0.647 | 0.659 | **0.675** | **0.708** |

## E   ADDITIONAL METRICS

We provide additional metrics for the label model comparisons shown in Table 2 in the main paper. Again, results are averaged over 4 random runs. Table 13 shows the weighted F1 score, an average over all classes weighted by the support of each class. Table 12 shows weighted mean average precision, a metric that summarizes the precision-recall curve across all classes. We compute the average precision individually for each class (one vs. rest) and then aggregate the scores by summing them weighted by the support of each class to produce the weighted mean average precision score.

Table 13: Weighted F1 score of various label models on training samples with at least one LF vote. The F1 is computed separately for each class and then averaged weighted by the support of each class. We highlight the best result in **blue** and the second best result in **bold**.

| Dataset | MV | DawidSkene | MeTaL | FS | Snorkel | WSGAN-Vector | WSGAN-Encoder |
|---|---|---|---|---|---|---|---|
| AwA2 - A | 0.641 | **0.665** | 0.62 | 0.619 | 0.636 | 0.637 | **0.684** |
| AwA2 - B | 0.604 | **0.661** | 0.58 | 0.597 | 0.593 | **0.664** | **0.672** |
| DomainNet | 0.603 | **0.655** | 0.443 | 0.622 | 0.468 | **0.654** | 0.634 |
| MNIST | 0.756 | 0.716 | 0.746 | 0.755 | 0.746 | **0.764** | **0.795** |
| FashionMNIST | 0.706 | 0.691 | 0.698 | 0.705 | 0.698 | **0.710** | **0.715** |
| GTSRB | **0.802** | 0.616 | 0.800 | 0.628 | 0.799 | **0.801** | **0.811** |
| CIFAR10-A | 0.824 | **0.850** | 0.797 | 0.796 | 0.798 | **0.851** | **0.872** |
| CIFAR10-B | 0.712 | 0.672 | 0.702 | 0.703 | 0.702 | **0.720** | **0.727** |
| CIFAR10-C | 0.726 | **0.741** | 0.723 | 0.713 | 0.718 | **0.738** | **0.759** |
| CIFAR10-D | 0.809 | **0.839** | 0.791 | 0.784 | 0.781 | 0.834 | **0.844** |
| CIFAR10-E | 0.864 | **0.901** | 0.843 | 0.825 | 0.839 | **0.899** | **0.916** |
| CIFAR10-F | 0.684 | 0.609 | 0.677 | 0.675 | 0.674 | **0.688** | **0.699** |

# F  THEORETICAL JUSTIFICATION

We provide additional setup and proofs for our two theoretical claims.

## F.1  CLAIM (1)

Our goal is to derive a generalization bound; that is, an upper bound on $|\hat{\mathbb{R}}_{\hat{\mathcal{D}}} - \mathbb{R}_{\mathcal{D}}|$. In words, this is the gap between the loss on a sample drawn from the true distribution and the empirical loss we obtained by training on the weakly-supervised dataset with unlabeled data sampled from the generative model.

**Mixture of Gaussians**  Recall that $D$ is the joint distribution of the unlabeled and labeled points. Let's call the unlabeled data marginal distribution $D_X$. Then, we make the assumption that $D_X$ is a mixture of $k$ Gaussians. Here, there is some relationship between the mixtures and the two classes, but we need not further specify it. Using the result (Ashtiani et al., 2018), we get that the number of samples needed to learn $D_X$ up to $\varepsilon$ in total variation distance is $\tilde{\Theta}(kd^2/\varepsilon^2)$.

Note that in fact this expression hides some polylogarithmic terms. However, for simplicity, we're going to ignore these terms and just pretend that the necessary bound is $c_G kd^2/\varepsilon^2$, where $c_G$ is some constant for learning a density.

Based on this, we'll make the following assumption. We perform density estimation on $n_1$ samples from $D_X$ and obtain some model $g$ such that distribution of $g$ (we'll abuse notation and just refer to this as the model itself $g$) and $D_X$ satisfies

$$d_{\text{TV}}(D_X, g) \leq d\sqrt{\frac{c_G k}{n_1}}. \tag{5}$$

So now we have control over one marginal (the unlabeled data). Let's work on the conditional term next.

**Majority Vote**  For simplicity, let's assume that we use majority vote as the aggregation scheme for the $m$ labeling functions. We make the following assumptions. The labeling functions have accuracy $1/2 + \alpha$, for some $\alpha \in (0, 1/2]$, in the following sense. For any datapoint $(X, Y)$, the probability of a labeling function guessing the value of $Y$ correctly is $1/2 + \alpha$, and the probability of any guessing wrong is $1/2 - \alpha$. This holds for all values of $X$. Note: these are very strong assumptions.

The probability that we make a mistake, e.g., that majority vote aggregates votes to 0 when $Y = 1$ or vice-versa is given by the binomial CDF $F(m/2, m, \alpha + 1/2)$, which has the following simple bound that follows from Hoeffding's inequality,

$$F(m/2, m, \alpha + 1/2) \leq \exp\left(-2m\left(\alpha + 1/2 - \frac{m/2}{m}\right)^2\right) = \exp(-2m\alpha^2).$$

With the above, as $D_{Y|X}$ is a Bernoulli random variable, we can directly upper bound the total variation distance between $D_{Y|X}$ and $D_{\hat{Y}|X}$:

$$d_{\text{TV}}(D_{Y|X}, D_{\hat{Y}|X}) \leq \exp(-2m\alpha^2). \tag{6}$$

**Joint Distribution**  Now we have some control over the generative model's error (from the density estimation bound) and some control over the label recovery (from the above bound resulting from majority vote). Now we put it together. First, we write down some useful inequalities between the total variation distance and the Hellinger distance (Duchi, 2016) (Prop 2.10). These are, for densities $p, q$,

$$D_{\text{hel}}(p, q) \leq \sqrt{2d_{\text{TV}}(p, q)} \tag{7}$$

and

$$d_{\text{TV}}(p, q) \leq D_{\text{hel}}(p, q)\sqrt{1 - D_{\text{hel}}(p, q)^2/4}. \tag{8}$$

We use $p$ as the density for $\mathcal{D}$ and $q$ as the density for $\hat{\mathcal{D}}$, and write $p = p_1(x)p_2(y|x)$, $q = q_1(x)q_2(y|x)$. First, using Eq. (5) and Eq. (7), we have that

$$D_{\text{hel}}(\mathcal{D}_X, g) \le \sqrt{2d_{\text{TV}}(\mathcal{D}_X, g)} \le \left(\frac{4c_G k d^2}{n_1}\right)^{\frac{1}{4}}. \tag{9}$$

Then, using Eq. (6) and Eq. (7), we get

$$D_{\text{hel}}(D_{Y|X}, D_{\hat{Y}|X}) \le \sqrt{2d_{\text{TV}}(D_{Y|X}, D_{\hat{Y}|X})} \le \sqrt{2\exp(-2m\alpha^2)} = \sqrt{2}\exp(-m\alpha^2). \tag{10}$$

Next,

$$\begin{aligned}
D_{\text{hel}}(\mathcal{D}, \hat{\mathcal{D}}) &= \int\int (\sqrt{p_1(x)p_2(y|x)} - \sqrt{q_1(x)Q_2(y|x)})^2 dy dx \\
&= \int\int \left(p_1(x)p_2(y|x) + q_1(x)q_2(y|x) - 2\sqrt{p_1(x)p_2(y|x)q_1(x)q_2(y|x)}\right) dy dx \\
&= 2 - 2\int\int \sqrt{p_1(x)p_2(y|x)q_1(x)q_2(y|x)} dy dx \\
&= 2 - 2\int \sqrt{p_1(x)q_1(x)} \left(1 - \frac{1}{2}D_{\text{hel}}(p_2, q_2)\right) dx \\
&\le 2 - 2\int \sqrt{p_1(x)q_1(x)} \left(1 - \frac{\sqrt{2}}{2}\exp(-m\alpha^2)\right).
\end{aligned}$$

Note that here, we use the fact that our bound holds for all conditional distributions regardless of $x$. Continuing,

$$\begin{aligned}
D_{\text{hel}}(\mathcal{D}, \hat{\mathcal{D}}) &\le 2 - 2\int \sqrt{p_1(x)q_1(x)} \left(1 - \frac{\sqrt{2}}{2}\exp(-m\alpha^2)\right) \\
&= 2 - 2(1 - \frac{1}{2}D_{\text{hel}}(p_1, q_1)) \left(1 - \frac{\sqrt{2}}{2}\exp(-m\alpha^2)\right) \\
&\le 2 - 2\left(1 - \frac{1}{2}\left(\frac{4c_G k d^2}{n_1}\right)^{\frac{1}{4}}\right) \left(1 - \frac{\sqrt{2}}{2}\exp(-m\alpha^2)\right) \\
&= \left(\frac{4c_G k d^2}{n_1}\right)^{\frac{1}{4}} + \sqrt{2}\exp(-m\alpha^2) - \left(\frac{c_G k d^2}{n_1}\right)^{\frac{1}{4}}\exp(-m\alpha^2).
\end{aligned}$$

Now we apply Eq. (8) to get the bound back into the total variation distance setting. We have

$$\begin{aligned}
d_{\text{TV}}(\mathcal{D}, \hat{\mathcal{D}}) &\le D_{\text{hel}}(\mathcal{D}, \hat{\mathcal{D}})\sqrt{1 - D_{\text{hel}}(\mathcal{D}, \hat{\mathcal{D}})^2/4} \le D_{\text{hel}}(\mathcal{D}, \hat{\mathcal{D}}) \\
&\le \left(\frac{4c_G k d^2}{n_1}\right)^{\frac{1}{4}} + \sqrt{2}\exp(-m\alpha^2) - \left(\frac{c_G k d^2}{n_1}\right)^{\frac{1}{4}}\exp(-m\alpha^2) \\
&\le \left(\frac{4c_G k d^2}{n_1}\right)^{\frac{1}{4}} + \sqrt{2}\exp(-m\alpha^2). \tag{11}
\end{aligned}$$

**Bounding the Risk** The final task is to bound the risk. First, suppose we are training a classifier chosen from a function class $\mathcal{F}$, trained on $n_2$ independently-drawn data points. Then, a standard result is that with probability at least $1 - \delta$,

$$\sup_{f \in \mathcal{F}} |\hat{\mathbb{R}}_{\mathcal{D}}(f) - \mathbb{R}_{\mathcal{D}}(f)| \le 2\mathfrak{R} + \sqrt{\frac{\log(1/\delta)}{2n_2}}. \tag{12}$$

Here, $\mathfrak{R}$ is the Rademacher complexity of the function class. However, the above is for training on samples from the true distribution. Instead, we can write

$$\begin{aligned}
|\hat{\mathbb{R}}_{\hat{\mathcal{D}}} - \mathbb{R}_{\mathcal{D}}| &= |\hat{\mathbb{R}}_{\hat{\mathcal{D}}} - \mathbb{R}_{\hat{\mathcal{D}}} + \mathbb{R}_{\hat{\mathcal{D}}} - \mathbb{R}_{\mathcal{D}}| \\
&\le |\hat{\mathbb{R}}_{\hat{\mathcal{D}}} - \mathbb{R}_{\hat{\mathcal{D}}}| + |\mathbb{R}_{\hat{\mathcal{D}}} - \mathbb{R}_{\mathcal{D}}|.
\end{aligned}$$

For the right-hand term, we have the following:

$$|\mathbb{R}_{\hat{\mathcal{D}}} - \mathbb{R}_{\mathcal{D}}| = |\int \ell(f(x), y)|p(x, y) - q(x, y)|d\mu$$

$$\leq B_\ell d_{\text{TV}}(\hat{\mathcal{D}}, \mathcal{D}).$$

Then, putting this together with the expression in Eq. (11) into Eq. (12), we get that, with probability at least $1 - \delta$,

$$\sup_{f \in \mathcal{F}} |\hat{\mathbb{R}}_{\hat{\mathcal{D}}}(f) - \mathbb{R}_{\mathcal{D}}(f)| \leq (\sup_{f \in \mathcal{F}} |\hat{\mathbb{R}}_{\hat{\mathcal{D}}} - \mathbb{R}_{\hat{\mathcal{D}}}| + |\mathbb{R}_{\hat{\mathcal{D}}} - \mathbb{R}_{\mathcal{D}}|)$$

$$\leq 2\mathfrak{R} + \sqrt{\frac{\log(1/\delta)}{2n_2}} + B_\ell d_{\text{TV}}(\hat{\mathcal{D}}, \mathcal{D})$$

$$\leq 2\mathfrak{R} + \sqrt{\frac{\log(1/\delta)}{2n_2}} + B_\ell \left(\frac{4c_G k d^2}{n_1}\right)^{\frac{1}{4}} + B_\ell \sqrt{2} \exp(-m\alpha^2). \quad (13)$$

**Interpreting the Bound**  In Eq. (13), we saw that

$$\sup_{f \in \mathcal{F}} |\hat{\mathbb{R}}_{\hat{\mathcal{D}}}(f) - \mathbb{R}_{\mathcal{D}}(f)| \leq 2\mathfrak{R} + \sqrt{\frac{\log(1/\delta)}{2n_2}} + B_\ell \left(\frac{4c_G k d^2}{n_1}\right)^{\frac{1}{4}} + B_\ell \sqrt{2} \exp(-m\alpha^2).$$

Now let's interpret this result piece-by-piece. The terms are the following

- The Rademacher complexity of the function class, which is present in the standard generalization bound.
- An estimation error term as a function of how much data we have to train our classifier $n_2$. It has the standard rate $1/\sqrt{n_2}$. Again, this is standard in any bound.
- A penalty term due to the generative model usage. It tells us how much we lose by training on generated data rather than (unlabeled) data from the true distribution. It scales as $n_1^{-1/4}$, where $n_1$ is the number of samples of unlabeled data used to train the generative model. Note also the dependence on the number of mixture components and dimension.
- A penalty term due to weak supervision. It tells us what we lose by using estimated (pseudo)labels rather than true labels; we note that the penalty scales exponentially in the number of labeling functions $m$, but is slowed down by small $\alpha$, as our accuracies are $\alpha$ better than random.

### F.2  CLAIM (2)

Our proof of claim (2) uses the setting of Thekumparampil et al. (2018), which introduces RCGAN. RCGAN is a conditional GAN architecture that corrupts the label before passing them to the discriminator by passing the true labels through a noisy channel. The authors provide a multiplicative approximation bound between the GAN loss under the unobserved true labels and the loss under the noisy labels. This noisy channel model acts as a nice model of the label generating process of weak supervision. Using this noisy channel model, we can control the amount of label corruption to match that of weak supervision.

Following the setup of Thekumparampil et al. (2018), we define a function that multiplies a one-hot encoded true label vector by a right-stochastic matrix $C \in \mathbb{R}^{2 \times 2}$ where $C_{i,j} = P(\tilde{y}_j | y_i)$—this is our noisy channel. This induces a joint distribution $\widetilde{P}_{X, \tilde{Y}}$ for the examples $x$ and noisy labels $\tilde{y}$ from the conditional distribution defined by $C$. We restate the theorems of interest from Thekumparampil et al. (2018) here, and proceed to adapt them to our problem setting.

**Theorem 1.** *(Multiplicative bound on the total variation distance from Thekumparampil et al. (2018).) Let $P_{X,Y}$ and $Q_{X,Y}$ be two distributions over $\mathcal{X} \times \{0, 1\}$ and let $\widetilde{P}_{X, \tilde{Y}}$ and $\widetilde{Q}_{X, \tilde{Y}}$ be the corresponding distributions with noisy labels from $C$. If $C$ is full-rank, then*

$$d_{TV}(\widetilde{P}, \widetilde{Q}) \leq d_{TV}(P, Q) \leq \|C^{-1}\|_\infty \, d_{TV}(\widetilde{P}, \widetilde{Q}). \quad (14)$$

Theorem 1 says that the total variation distance between the true noisy distribution and the noisy generated distribution from RCGAN approximate its noiseless counterpart up to a factor of $\|C^{-1}\|_\infty$. Our goal is to construct $C_{\epsilon_{MV}}$ to model the noise from weak supervision (in particular, majority vote) and show that it leads to a tighter bound than when we directly plug in the labels from a single LF into Theorem 1. To begin, consider the following parameterization of $C$, with $\epsilon \in (0, 1/2)$:

$$C_\epsilon = I_2 + \begin{bmatrix} -\epsilon & \epsilon \\ \epsilon & -\epsilon \end{bmatrix} = \begin{bmatrix} 1-\epsilon & \epsilon \\ \epsilon & 1-\epsilon \end{bmatrix}. \tag{15}$$

Here, $\epsilon$ denotes the labeling error for each class. Given this parameterization, we obtain the following expression for $\|C_\epsilon^{-1}\|_\infty$.

$$\|C_\epsilon^{-1}\|_\infty = \left\| \begin{bmatrix} 1-\epsilon & \epsilon \\ \epsilon & 1-\epsilon \end{bmatrix}^{-1} \right\|_\infty \tag{16}$$

$$= |((1-\epsilon)^2 - \epsilon^2)^{-1}| \left\| \begin{bmatrix} 1-\epsilon & -\epsilon \\ -\epsilon & 1-\epsilon \end{bmatrix} \right\|_\infty \tag{17}$$

$$= (1 - 2\epsilon)^{-1} \tag{18}$$

Note that $C_\epsilon$ is full-rank as it has a finite inverse. It is also clear that $\|C_\epsilon^{-1}\|_\infty$ is a monotonically increasing function of $\epsilon$. That is to say that if we do something to decrease the labeling error $\epsilon$, then $\|C_\epsilon^{-1}\|_\infty$ also decreases and we obtain a tighter bound. We will go on to derive an expression for the labeling error under majority vote with $m$ LFs, $\epsilon_{MV}$, and show that it is smaller than the labeling error from a single LF, $\epsilon_\lambda$. Namely, we want find a condition where $\epsilon_{MV} \leq \epsilon_\lambda$ holds and that majority vote leads to an improved Theorem 1 bound.

**Proposition 1.** *(Total variation version.) Let $\epsilon_{MV}$ be the labeling error from majority vote from $m$ LFs, where $m \geq \frac{\log(1/\epsilon_\lambda)}{2\left(\frac{1}{2} - \epsilon_\lambda\right)^2}$, whose individual labeling errors are each $\epsilon_\lambda$. Then the following holds*

$$d_{TV}(\widetilde{P}_{MV}, \widetilde{Q}_{MV}) \leq d_{TV}(P, Q) \leq \|C_{\epsilon_{MV}}^{-1}\|_\infty \, d_{TV}(\widetilde{P}_{MV}, \widetilde{Q}_{MV}) \leq \|C_{\epsilon_\lambda}^{-1}\|_\infty \, d_{TV}(\widetilde{P}_{MV}, \widetilde{Q}_{MV}).$$

*Proof.* We begin by deriving an upper bound on $\epsilon_{MV}$. We have LFs $\{\lambda_i\}_{i=1}^m$ that each produce incorrect predictions with probability $\epsilon_\lambda = \frac{1}{2} - \alpha$, using $\alpha$ as defined in Claim (1). Now, we need to show that the probability of producing incorrect predictions using majority vote with more label functions, $\{\lambda_i\}_{i=1}^m$, has error $\epsilon_{MV} \leq \epsilon_\lambda$. Define the event that $\lambda_i$ is incorrect as follows: $z_i = \mathbb{I}[\lambda_i \neq y]$, then $\mathbb{E}[[]\, z_i] = \epsilon_\lambda$. Using this, we apply Hoeffding's bound to $\epsilon_{MV}$.

$$\epsilon_{MV} = P\left( \sum_{i=1}^m z_i - m\epsilon_\lambda \geq \frac{m}{2} - m\epsilon_\lambda \right) \tag{19}$$

$$\leq \exp\left( \frac{-2\left(\frac{m}{2} - m\epsilon_\lambda\right)^2}{m} \right) \tag{20}$$

$$= \exp\left( -2m\left(\frac{1}{2} - \epsilon_\lambda\right)^2 \right). \tag{21}$$

Next, we plug the bound from (21) into (18) to obtain the following expression for $\|C_{\epsilon_{MV}}^{-1}\|_\infty$.

$$\|C_{\epsilon_{MV}}^{-1}\|_\infty = (1 - 2\epsilon_{MV})^{-1} \tag{22}$$

$$\leq \left( 1 - 2\exp\left( -2m\left(\frac{1}{2} - \epsilon_\lambda\right)^2 \right) \right)^{-1}. \tag{23}$$

To complete the proof, we need the following to hold: $\|C_{\epsilon_{MV}}^{-1}\|_\infty \leq \|C_{\epsilon_\lambda}^{-1}\|_\infty$, but due to the monotonicity of $\|C_\epsilon^{-1}\|_\infty$, it is sufficient to show that $\epsilon_{MV} \leq \epsilon_\lambda$.

Recall that $\epsilon_{MV} \leq \exp\left( -2m\left(\frac{1}{2} - \epsilon_\lambda\right)^2 \right)$, so if we set $\exp\left( -2m\left(\frac{1}{2} - \epsilon_\lambda\right)^2 \right) \leq \epsilon_\lambda$, we obtain the minimum number of label functions, $m$, required to ensure $\epsilon_{MV} \leq \epsilon_\lambda$.

$$\exp\left(-2m\left(\frac{1}{2}-\epsilon_\lambda\right)^2\right) \le \epsilon_\lambda$$

$$\Rightarrow m \ge \frac{\log(1/\epsilon_\lambda)}{2\left(\frac{1}{2}-\epsilon_\lambda\right)^2}.$$

Plugging (23) into Theorem 1, we obtain the following

$$d_{\mathrm{TV}}(\widetilde{P}_{\mathrm{MV}},\widetilde{Q}_{\mathrm{MV}}) \le d_{\mathrm{TV}}(P,Q) \le \|C_{\epsilon_{\mathrm{MV}}}^{-1}\|_\infty \; d_{\mathrm{TV}}(\widetilde{P}_{\mathrm{MV}},\widetilde{Q}_{\mathrm{MV}})$$

$$\le \left(1 - 2\exp\left(-2m\left(\frac{1}{2}-\epsilon_\lambda\right)^2\right)\right)^{-1} d_{\mathrm{TV}}(\widetilde{P}_{\mathrm{MV}},\widetilde{Q}_{\mathrm{MV}})$$

$$\le (1 - 2\epsilon_\lambda)^{-1} \, d_{\mathrm{TV}}(\widetilde{P}_{\mathrm{MV}},\widetilde{Q}_{\mathrm{MV}})$$

$$= \|C_{\epsilon_\lambda}^{-1}\|_\infty \; d_{\mathrm{TV}}(\widetilde{P}_{\mathrm{MV}},\widetilde{Q}_{\mathrm{MV}})$$

which completes the proof. $\qquad\square$

Notice that the proof of Proposition 1 does not depend on total variation distance beyond the dependence on Theorem 1. As such, Proposition 1 can be stated more generally in terms of the Integral Probability Metric induced by the GAN discriminator $\mathcal{F}$ using Theorem 2 of Thekumparampil et al. (2018):

$$d_{\mathcal{F}}(\widetilde{P}_{\mathrm{MV}},\widetilde{Q}_{\mathrm{MV}}) \le d_{\mathcal{F}}(P,Q) \le \|C_{\epsilon_{\mathrm{MV}}}^{-1}\|_\infty \; d_{\mathcal{F}}(\widetilde{P}_{\mathrm{MV}},\widetilde{Q}_{\mathrm{MV}}) \le \|C_{\epsilon_\lambda}^{-1}\|_\infty \; d_{\mathcal{F}}(\widetilde{P}_{\mathrm{MV}},\widetilde{Q}_{\mathrm{MV}}).$$

Finally, notice that Proposition 1 is made in terms of $d_{\mathcal{F}}(\widetilde{P}_{\mathrm{MV}},\widetilde{Q}_{\mathrm{MV}})$ and not in terms of $d_{\mathcal{F}}(\widetilde{P}_\lambda,\widetilde{Q}_\lambda)$. We can show that as the number of LFs approach infinity, we recover the distance under the clean labels: $d_{\mathcal{F}}(P,Q)$. Applying Theorem 1 to majority vote and a single LF results in the following two expressions:

$$d_{\mathcal{F}}(\widetilde{P}_{\mathrm{MV}},\widetilde{Q}_{\mathrm{MV}}) \le d_{\mathcal{F}}(P,Q) \le \|C_{\epsilon_{\mathrm{MV}}}^{-1}\|_\infty \; d_{\mathcal{F}}(\widetilde{P}_{\mathrm{MV}},\widetilde{Q}_{\mathrm{MV}}) \tag{24}$$

$$d_{\mathcal{F}}(\widetilde{P}_\lambda,\widetilde{Q}_\lambda) \le d_{\mathcal{F}}(P,Q) \le \|C_{\epsilon_\lambda}^{-1}\|_\infty \; d_{\mathcal{F}}(\widetilde{P}_\lambda,\widetilde{Q}_\lambda) \tag{25}$$

Rearranging terms, we obtain the following

$$1 \le \frac{d_{\mathcal{F}}(P,Q)}{d_{\mathcal{F}}(\widetilde{P}_{\mathrm{MV}},\widetilde{Q}_{\mathrm{MV}})} \le \|C_{\epsilon_{\mathrm{MV}}}^{-1}\|_\infty \le \|C_{\epsilon_\lambda}^{-1}\|_\infty$$

and

$$1 \le \frac{d_{\mathcal{F}}(P,Q)}{d_{\mathcal{F}}(\widetilde{P}_\lambda,\widetilde{Q}_\lambda)} \le \|C_{\epsilon_\lambda}^{-1}\|_\infty.$$

Notice that $\|C_{\epsilon_\lambda}^{-1}\|_\infty$ has no dependence on $m$ since it's a single LF, but $\|C_{\epsilon_{\mathrm{MV}}}^{-1}\|_\infty$ approaches 1 as $m \to \infty$:

$$\lim_{m\to\infty} 1 \le \frac{d_{\mathcal{F}}(P,Q)}{d_{\mathcal{F}}(\widetilde{P}_{\mathrm{MV}},\widetilde{Q}_{\mathrm{MV}})} \le \|C_{\epsilon_{\mathrm{MV}}}^{-1}\|_\infty \le \|C_{\epsilon_\lambda}^{-1}\|_\infty$$

$$\Rightarrow 1 \le \frac{d_{\mathcal{F}}(P,Q)}{d_{\mathcal{F}}(\widetilde{P}_{\mathrm{MV}},\widetilde{Q}_{\mathrm{MV}})} \le 1 \le \|C_{\epsilon_\lambda}^{-1}\|_\infty$$

$$\Rightarrow d_{\mathcal{F}}(P,Q) = d_{\mathcal{F}}(\widetilde{P}_{\mathrm{MV}},\widetilde{Q}_{\mathrm{MV}}).$$

Hence we obtain a stronger bound as the number of LFs increases.

## F.3 EXTENSIONS

For the sake of clarity, we make several simplifying assumptions in Claims (1) and (2). Both claims use the simplest possible aggregation strategy for weak supervision—majority vote, and our analysis in Claim (1) involves the use of a Gaussian mixture model—a less complex object of study compared to a GAN. We can directly extend both analyses to use more sophisticated weak supervision label models instead of majority vote, and different generative models, which should lead to improved bounds at the expense of a more complex claim statement. We note, additionally, that neither of the claims attempt to provide deep insight into the benefits of *jointly* learning the generative model and the label model—but this can be done with a slightly more careful analysis.

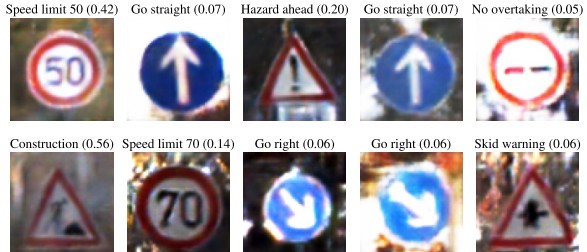

Figure 10: Images and pseudolabels generated by the proposed WSGAN (with a simple DCGAN architecture) on the weakly supervised GTSRB dataset. WSGAN can estimate labels even for images where no weak supervision sources provide information (see end of Section 3.1).

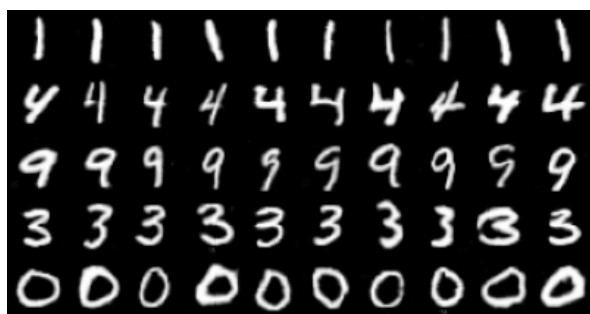

Figure 11: A random set of MNIST images generated by WSGAN-E (using a DCGAN), with the discrete latent random variable kept fix for each row of images.

## G   ADDITIONAL IMAGES

Here we provide additional generated images in Figures 10, 11, 12, 13, and 14. These random images are generated by WSGAN with a DCGAN base architecture, where the discrete latent variable $d$ passed to the generator is kept the same in each row.

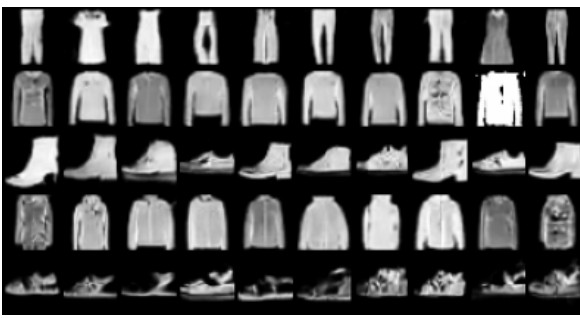

Figure 12: A random set of FashionMNIST images generated by WSGAN-E (using a DCGAN), with the discrete latent random variable kept fix for each row of images.

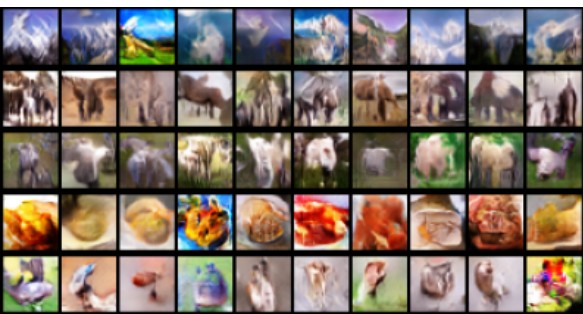

Figure 13: A random set of Domainnet images generated by WSGAN-E (using a DCGAN), with the discrete latent random variable kept fix for each row of images. Note that this dataset is particularly challenging for a GAN as our dataset has fewer than 7,000 images, resulting in considerably lower quality of synthetic images compared to GTSRB for example.

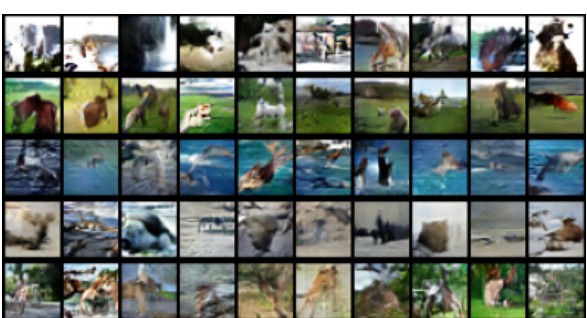

Figure 14: A random set of AwA2 images generated by WSGAN-E (using a DCGAN), with the discrete latent random variable kept fix for each row of images. Note that this dataset is particularly challenging for a GAN, as this weakly supervised dataset has fewer than 7,000 images, resulting in considerably lower quality of synthetic images compared to GTSRB for example.

