# OpenReview forum: "Generative Modeling Helps Weak Supervision (and Vice Versa)"
_ICLR.cc/2023/Conference — ICLR 2023 poster_

### Official Review · Reviewer_vpDG · 2022-10-25

**Confidence:** 3
**Clarity, Quality, Novelty And Reproducibility:** 1) Clarity. The paper is well written…
**Correctness:** 3
**Technical Novelty And Significance:** 3
**Empirical Novelty And Significance:** 3
**Recommendation:** 6

**Strength And Weaknesses:**

Strengths
1) The idea of fusing InfoGAN that discovers discrete latent factors and programmatic weak supervision based label model is interesting, and could also be very helpful for many practical tasks.
2) The experimental results are comprehensive and convincing. A considerable number of experiments are conducted to demonstrate the model's effectiveness on both weak supervision labeling and generative modeling, in addition to the ability of generating samples with corresponding label estimates.
3) Theoretical analyses are presented to support the claim that generative model and weak-supervision label model can help each other.

Weaknesses
1) Theoretical statement of Claim 2 is not precise enough lacking necessary explanation for related assumptions and analyses.
2) The descriptions to Table 4 is confusing. It is stated that an improvement in test accuracy up to 3.9% (or a modest increase of up to 2.4% later) was achieved. But there are many different datasets, why you only focus on the performance changes on one dataset?
3) Although the paper has demonstrated that the proposed method is the first to enable pseudo-labeled data augmentation, the idea of generating synthetic images with pseudo-labels is not novel in current generative models (e.g. GMM based deep generative models). And more experiments need to be conducted to demonstrate better qualities of generated pseudo-labeled samples compared to other generative methods.


**Summary Of The Paper:**

To tackle the problem of lacking sufficient ground-truth labels in practical applications, this paper proposed a paradigm to fuse the generative model and programmatic weak supervision method, which are often used separately to tackle this issue in previous literatures. The fusion is mainly achieved by encouraging an agreement between the prediction label from the weak-supervision label model and the inferred discrete latent code discovered with the InfoGAN. The agreement essentially means the labels predicted by the weak-supervision label model and InfoGAN are the same up to some permutation.

The authors demonstrate that the proposed fusion and alignment are technically solid, by theoretically showing that: 1) the loss gap between the original dataset and augmented dataset with generated samples is controlled by penalty from labeling function accuracy, indicating positive benefits of weak supervision; 2) the RCGAN loss with noisy labels generated by multiple labeling functions has a tighter bound than that by a single labeling function. And these two theoretical results together make it possible for weak supervision and generative modeling to cooperate each other.

Experiments are conducted to investigate the impacts on performance from four aspects: 1) label model accuracy; 2) classifier output and label model output alignment; 3) image generation quality; 4) augmented images by generator. Reported results show considerable performance boost for both InfoGAN and label model components as well as better alignment between their outputs than vanilla InfoGAN. Moreover, the proposed WSGAN enables pseudo-labeled samples generation with trained generator and classifier / label model, allowing for augmented training dataset without extra data collection or labeling to improve the downstream classifier performance.


**Summary Of The Review:**

The paper is generally clearly written and provides a novel perspective to incorporate weak supervision into generative modeling framework. Extensive experiments are conducted to demonstrate the effectiveness of the proposed fusion approach.

---

> ### Author Response · Authors · 2022-11-12
> **Response to Reviewer vpDG**
>
> Thank you for the feedback and suggestions. Below, we respond to the points you raised.
>
> We would also like to highlight that we have updated our submission with additional experiments on higher resolution images (256 by 256 pixels) of the LSUN scene category dataset. The results can be found in the network ablation in Section 4.2.3 and generated images in the Appendix Fig. 8.
>
> We use blue text in our pdf submission to highlight the newest changes that were made in response to the reviews.
>
> > Theoretical statement of Claim 2 is not precise enough
>
> Thank you for pointing out that Claim 2 requires more clarification. We provide the formal statement and proof of Claim 2 in Appendix F (pages 28-30). We have now added additional clarifications about the distributions and noise sources involved in Claim 2 to the main paper—we hope that this improves the presentation of this claim.
>
> > The descriptions to Table 4 is confusing.
>
> Thank you for the feedback. We have updated this part in our submission to reflect the results achieved across all datasets more broadly.
>
> > … the idea of generating synthetic images with pseudo-labels is not novel in current generative models (e.g. GMM based deep generative models). And more experiments need to be conducted to demonstrate better qualities of generated pseudo-labeled samples compared to other generative methods.
>
> The key difference between our work and prior synthetic image+label efforts is that we only require weak supervision sources, without any knowledge of their accuracies or levels of noise, and with no ground truth labels whatsoever. This way, a weakly supervised training pipeline can be augmented further with synthetic data. In contrast, to the best of our knowledge, most existing studies have focused on settings where there exist ground-truth labels, and have used them in e.g. conditional GANs or conditional VAEs.

---

### Official Review · Reviewer_ERZw · 2022-10-26

**Confidence:** 4
**Correctness:** 4
**Technical Novelty And Significance:** 3
**Empirical Novelty And Significance:** 3
**Recommendation:** 6

**Clarity, Quality, Novelty And Reproducibility:**

Clarity: The paper reads well enough, but it took me some time to figure out how the generative setup could be used for inference as well. Perhaps that ought to be emphasized for clarity

Quality and Novelty: I felt that the use of the disentangling GAN setup to be an interesting, novel approach and the results are convincing. However, it seems to me that some more insight into what the latent space learns (as we are seemingly producing a discrete label) through some visuals would be illuminating.

Reproducibility: I think the algorithmic components are properly laid out. However, I am not sure how easy it is to reproduce the experiments and whether the training dynamics are stable - especially, across the many datasets and experiments that the paper reports. Can the authors comment on this aspect?



**Strength And Weaknesses:**

+ The use of generative modeling with a disentangling setup like InfoGAN is a creative way to generate pseudolabels.
+ The evaluations show clear improvements over other approaches
- Failure cases for GAN generations not discussed.
- I also think that the applicability of the approach might be slightly limited in other areas such as speech, or NLP.
- Some visuals of what the latent space is learning would be illuminating.

**Summary Of The Paper:**

This paper proposes a generative modeling setup for weak supervision in images. The claim is that by doing so, there are improvements both to generative modeling and to the pseudolabeling qualities of the model, and they work in concert. They do this through proofs and experimental evaluations.

The setup is a GAN architecture based on InfoGAN, with hooks added for weak supervision. The InfoGAN latent code c is to be used as pseudolabel, trained by 'aligning' with the label generated by the labeling model LM through a loss construction. During training, c is produced by the synthetic image produced by the generator. However, during inference, one passes an actual test image to the encoder Q. If the GAN is trained properly, it is assumed that Q will work with 'real' images as well.

For evaluations, the authors show improvements as measured in FID (for image quality improvements) and posterior accuracy, F1 score (for labeling model) over a number of datasets, tasks and competitive models.

**Summary Of The Review:**

The paper presents an approach to add weak supervision to GAN modeling, and shows that the addition is useful for both pseudo-labeling and generative modeling. The presentation is clear, and results seem convincing.

---

> ### Author Response · Authors · 2022-11-12
> **Response to Reviewer ERZw**
>
> Thank you for your feedback! Below, we respond to the points you raised.
>
> We would also like to highlight that we have updated our submission with additional experiments on higher resolution images (256 by 256 pixels) of the LSUN scene category dataset. The results can be found in the network ablation in Section 4.2.3 and generated images in the Appendix Fig. 8.
>
> We use blue text in our pdf submission to highlight the newest changes that were made in response to the reviews.
>
> >  the applicability of the approach might be slightly limited in other areas such as speech, or NLP.
>
> We believe that our approach is also applicable to different data modalities with a modality specific GAN design. However, since generative models for modalities like image/sound/text currently follow widely different approaches, we chose to focus on images for this paper. We plan to explore other modalities in the future.
>
> > Some visuals of what the latent space is learning would be illuminating.
>
> When we visualize generated images in our paper, the latent code is fixed in each row. So for the different datasets, Figures 1,6, 8-14 show images generated based on different values of the sampled noise variable, while the discrete code remains fixed. This provides a visual sample for the conditional distribution for each class / discrete latent code.
>
> > Failure cases for GAN generations not discussed.
>
> Thank you for pointing this out. We have added some discussion on this to our conclusion and to our model training section in the appendix.
>
> > reproducibility
>
> We will make all our code to reproduce experiments open source, including the code to use a StyleGAN together with WSGAN. We will also release the weak labels for all datasets used in our paper.
>
> > I am not sure … whether the training dynamics are stable - especially, across the many datasets and experiments that the paper reports. Can the authors comment on this aspect?
>
> Thank you for this question. We have now added some clarifications to our discussion on WSGAN training and failure cases in the Appendix, and respond in detail below.
>
> In our experiments on smaller images, using DCGAN networks, we found WSGAN—while still sensitive to hyperparameter settings just as the GAN networks it is based on—to be more stable than training networks with a discrete latent code that use no weak supervision or ground-truth during training (i.e. InfoGAN). So the stability of training dynamics when discrete latent variables were involved **improved** with the use of weak supervision, despite the high levels of noise in the weakly supervised datasets we created.
>
> For our StyleWSGAN network ablations, we observed stable training comparable to the unconditional StyleGAN2-ADA with the simple adjustments described next. As we note in Appendix A (3rd paragraph), our experiments with a StyleGAN2 base network showed that the stability of the training dynamics was influenced by the size of the discrete code embedding (i.e. the dimensionality the sampled discrete code is mapped to before being normalized and concatenated with the sampled noise) and the learning rates for the WSGAN terms.  For high levels of noise in the weak supervision sources, the size of the code embedding had to be reduced from its standard settings closer to the number of classes in order to obtain stable training dynamics. With more accurate and higher coverage weak supervision sources, larger embedding dimensions were possible and did not destabilize training. The stable settings for the learning rates for WSGAN terms were defined as a fraction of the base StyleGAN learning rate. We did not have to adjust them for different datasets: one setting worked for all. Finally, we find that reducing the depth of the style network from its standard 8 layers can destabilize training, and that StyleWSGAN had to be trained for fewer epochs than an unconditional StyleGAN2-ADA.

---

### Official Review · Reviewer_kuyc · 2022-10-28

**Confidence:** 4
**Correctness:** 4
**Technical Novelty And Significance:** 4
**Empirical Novelty And Significance:** 4
**Recommendation:** 8

**Clarity, Quality, Novelty And Reproducibility:**

As mentioned above, the paper is clear and well-written.
The tackled problem is novel and interesting to the community.
As code and model are not provided, reproducibility is in question and I am wondering if the authors have plans to release the code and the model.

**Strength And Weaknesses:**

The paper is well-written and straightforward to follow. Technical terminology is adequately provided and checking the mathematics for soundness is relatively easy. While the theoretical justification (bounding in the appendix) was not carefully checked, it looked very interesting and it is definitely helpful for interested readers.

Both weak supervision and generative model are significant to the community to reduce the huge cost of labeling data. Combining them to further improve the labeling process matters to both academic researchers and industrial practitioners. This interesting problem is well-studied in the paper and a powerful and effective solution (WSGAN) is provided and justified theoretically and empirically.

One weakness is that the generated images are mostly low-resolution. While StyleGAN architecture is tried, the results in Figure 8 are still too small. I am wondering if the authors have plans to generate more realistic images.

**Summary Of The Paper:**

The paper has tackled the problem of simultaneously dealing with generative modeling and weak supervision. The authors have claimed that the generative model and weak supervision mutually benefit each other. The proposed method (WSGAN) has improved weak supervision, generative modeling, and data augmentation.
Technical challenges arise from developing the interface between the label model (weak supervision) and the generative model and fusing them. The challenges are solved by the proposed WSGAN and theoretical justification is provided in addition to empirical evidence.

**Summary Of The Review:**

The paper is well-written and a significant problem is solved in a novel way.
The method is mathematically sound and comprehensive experiments are performed.

---

> ### Author Response · Authors · 2022-11-12
> **Response to Reviewer kuyc**
>
> Thank you for taking the time to review our submission. Please note that we use blue text in our pdf submission to highlight the newest changes that were made in response to the reviews.
>
> > One weakness is that the generated images are mostly low-resolution.
>
> We agree—since submission, we have updated our paper in Section 4.2.3 to include experiments with 256 by 256 pixel LSUN scene category images as part of our network ablation with StyleWSGAN (i.e. our WSGAN approach using a StyleGAN2-ADA). On the weakly supervised LSUN subset that we created, StyleWSGAN improves FID compared to the unconditional StyleGAN2-ADA . Some examples of generated images can be found in our Appendix in Fig. 8.
>
> The results demonstrate that the approach can scale to higher resolution images with the appropriate base network and that the use of weak supervision can improve image generation quality in these cases as well.

---

### Decision · Program_Chairs · 2023-01-20

**Decision:**

Accept: poster

**Justification For Why Not Higher Score:**

This is a good paper, but there wasn't overwhelming enthusiasm for this amongst the reviewers.

**Justification For Why Not Lower Score:**

This is a solid piece of work.

**Metareview: Summary, Strengths And Weaknesses:**


The paper discusses the relationship generative modeling and weak supervision. The paper is generally well written and has clear technical and empirical support. The discussion phase was useful in helping improve the paper, although the reviewer scores remained the same. In general I think ICLR readers will find this an interesting and accessible piece of work.


**Note From Pc:**

if the above contains the word "oral" or "spotlight" please see: "oral" presentation means -> notable-top-5% and "spotlight" means -> notable-top-25%. As stated in our emails, we are disassociating presentation type from AC recommendations